# Unique structure and positive selection promote the rapid divergence of *Drosophila* Y chromosomes

**Ching-Ho Chang**[1][*][†], **Lauren E Gregory**[1], **Kathleen E Gordon**[2][‡], **Colin D Meiklejohn**[2], **Amanda M Larracuente**[1][*]

[1]Department of Biology, University of Rochester, Rochester, United States; [2]School of Biological Sciences, University of Nebraska-Lincoln, Lincoln, United States

**\*For correspondence:**
cchang2@fredhutch.org (C-HC);
alarracu@UR.Rochester.edu
(AML)

**Present address:** [†]Division of
Basic Sciences, Fred Hutchinson
Cancer Research Center, Seattle,
United States; [‡]Department of
Molecular Biology and Genetics,
Field of Genetics, Genomics and
Development, Cornell University,
Ithaca, United States

**Competing interest:** The authors
declare that no competing
interests exist.

**Reviewing Editor:** Daniel R
Matute, University of North
Carolina, Chapel Hill, United
States

## Abstract

Y chromosomes across diverse species convergently evolve a gene-poor, heterochromatic organization enriched for duplicated genes, LTR retrotransposons, and satellite DNA. Sexual antagonism and a loss of recombination play major roles in the degeneration of young Y chromosomes. However, the processes shaping the evolution of mature, already degenerated Y chromosomes are less well-understood. Because Y chromosomes evolve rapidly, comparisons between closely related species are particularly useful. We generated de novo long-read assemblies complemented with cytological validation to reveal Y chromosome organization in three closely related species of the *Drosophila simulans* complex, which diverged only 250,000 years ago and share >98% sequence identity. We find these Y chromosomes are divergent in their organization and repetitive DNA composition and discover new Y-linked gene families whose evolution is driven by both positive selection and gene conversion. These Y chromosomes are also enriched for large deletions, suggesting that the repair of double-strand breaks on Y chromosomes may be biased toward microhomology-mediated end joining over canonical non-homologous end-joining. We propose that this repair mechanism contributes to the convergent evolution of Y chromosome organization across organisms.

## Editor's evaluation

This manuscript by Chang et al. reports the evolutionary patterns of Y-chromosome evolution in *Drosophila*, providing perhaps the most comprehensive interspecific comparison of Y chromosomes available to date. They focus on four species of the *melanogaster* species subgroup and do extensive sequencing and assembly. The manuscript describes the pattern of divergence in these chromosomes, and uses comparative approaches to compare the drivers of evolution in flies and mammals. The authors suggest that the Y chromosome uses a different mechanism to repair double strand breaks than on autosomes. We were impressed by the novelty and rigor of the work as well as the overall presentation of the results.

## Introduction

Most sex chromosomes evolved from a pair of homologous gene-rich autosomes that acquired sex-determining factors and subsequently differentiated. Y chromosomes gradually lose most of their genes, while their X chromosome counterparts tend to retain the original autosomal complement of genes. This Y chromosome degeneration follows a suppression of recombination (*Rice, 1987a*), which limits the efficacy of natural selection, and causes the accumulation of deleterious mutations through Muller's ratchet, background selection, and hitchhiking effects (*Bachtrog, 2013*; *Charlesworth, 1978*;

*Rice, 1987b*; *Charlesworth et al., 1995*; *Charlesworth and Charlesworth, 2000*). As a consequence, many Y chromosomes present a seemingly hostile environment for genes, with their mutational burden, high repeat content, and abundant silent chromatin.

Genomic studies of Y chromosome evolution focus primarily on young sex chromosomes, addressing how the suppression of recombination promotes Y chromosome degeneration at both the epigenetic and genetic levels (*Bachtrog, 2013*; *Bergero et al., 2015*). Although sexually antagonistic selection is traditionally cited as the cause of recombination suppression on the Y chromosome, direct evidence for its role is still lacking (*Bergero et al., 2019*) and new models propose that regulatory evolution is the initial trigger for recombination suppression (*Lenormand et al., 2020*). Regardless of its role in initiating recombination suppression, on degenerating Y chromosomes, sexually antagonistic selection may accelerate Y-linked gene evolution to optimize male-specific functions. Indeed, Y-linked genes tend to have slightly higher rates of protein evolution than their orthologs on other chromosomes (*Bachtrog, 2003*; *Singh et al., 2014*). Higher rates of Y-linked gene evolution are driven by positive selection, relaxed selective constraints and male-biased mutation patterns, with most Y-linked genes evolving under at least some functional constraint (*Singh et al., 2014*). Although there is evidence suggesting that some Y chromosomes have experienced recent selective sweeps (*Larracuente and Clark, 2013*; *Bachtrog, 2004*), the relative importance of positive selection in shaping Y chromosome evolution remains unclear.

Y chromosomes harbor extensive structural divergence between species, in part through the acquisition of genes from other genomic regions (*Soh et al., 2014*; *Rozen et al., 2003*; *Hughes and Page, 2015*; *Bachtrog et al., 2019*; *Tobler et al., 2017*; *Peichel et al., 2019*; *Brashear et al., 2018*; *Hall et al., 2016*). However, the functions of most Y-linked genes are unknown (*Tobler et al., 2017*; *Hall et al., 2016*; *Chang and Larracuente, 2019*; *Carvalho et al., 2015*). Some Y-linked genes are duplicated and, in extreme cases, amplified into so-called ampliconic genes—gene families with tens to hundreds of highly similar sequences. Y chromosomes of both *Drosophila* and mammals have independently acquired and amplified gene families, which turnover rapidly between closely related species (*Soh et al., 2014*; *Bachtrog et al., 2019*; *Brashear et al., 2018*; *Ellison and Bachtrog, 2019*; *Hughes et al., 2010*; *Mueller et al., 2008*). Following Y-linked gene amplification, gene conversion between gene copies may enhance the efficacy of selection on Y-linked genes in the absence of crossing over (*Rozen et al., 2003*; *Connallon and Clark, 2010*).

Detailed analyses of old Y chromosomes have been restricted to a few species with reference-quality assemblies, for example, mouse and human. The challenges of cloning and assembling repeat-rich regions of the genome have stymied progress towards a complete understanding of Y chromosome evolution (*Carlson and Brutlag, 1977*; *Lohe and Brutlag, 1987a*; *Lohe and Brutlag, 1987b*). Recent advances in long-read sequencing make it feasible to assemble large parts of Y chromosomes (*Peichel et al., 2019*; *Hall et al., 2016*; *Chang and Larracuente, 2019*; *Mahajan et al., 2018*) enabling comparative studies of a majority of Y-linked sequences in closely related species.

*Drosophila melanogaster* and three related species in the *D. simulans* clade are ideally suited to study Y chromosome evolution. These Y chromosomes are functionally divergent, contribute to hybrid sterility (*Araripe et al., 2016*; *Bayes and Malik, 2009*; *Johnson et al., 1992*; *Coyne, 1985*), and at least four X-linked meiotic drive systems likely shape Y chromosome evolution in these species (*Bozzetti et al., 1995*; *Courret et al., 2019*; *Tao et al., 2007*; *Tao et al., 2001*; *Helleu et al., 2019*; *Branco et al., 2013*; *Montchamp-Moreau et al., 2001*; *Meiklejohn et al., 2018*). Previous genetic and transcriptomic studies suggest that Y chromosome variation can impact male fitness and gene regulation (*Reijo et al., 1995*; *Vogt et al., 1996*; *Sun et al., 2000*; *Repping et al., 2003*; *Morgan and Pardo-Manuel de Villena, 2017*; *Lemos et al., 2010*; *Wang et al., 2018*; *Sackton et al., 2011*). Since there is minimal nucleotide variation and divergence in Y-linked protein-coding sequences within and between these *Drosophila* species (*Singh et al., 2014*; *Larracuente and Clark, 2013*; *Helleu et al., 2019*), structural variation may be responsible for the majority of these effects. For example, 20–40% of *D. melanogaster* Y-linked regulatory variation (YRV) comes from differences in ribosomal DNA (rDNA) copy numbers (*Zhou et al., 2012*). The chromatin on *Drosophila* Y chromosomes has genome-wide effects on expression level and chromatin states (*Brown and Bachtrog, 2017*), but aside from the rDNA, the molecular basis of Y chromosome divergence and variation in these species remains elusive.

**Table 1.** Contiguity statistics for heterochromatin-enriched assemblies.

| Y chromosome assembly | # of contigs | Total length | Contigs N50 |
|---|---|---|---|
| *D. melanogaster** | 80 | 14,578,684 | 416,887 |
| *D. mauritiana*† | 55 | 17,880,069 | 1,628,994 |
| *D. simulans*† | 38 | 13,717,056 | 1,031,383 |
| *D. sechellia*† | 63 | 14,899,148 | 555,130 |

*__*Chang and Larracuente, 2019__.
†This paper.

To study the factors and forces shaping the evolution of Y chromosome structure, we assembled the Y chromosomes of the three species in the *D. simulans* clade to reveal their structure and evolution and compared them to *D. melanogaster*. We find that the Y chromosomes of the *D. simulans* clade species have high duplication and gene conversion rates that, along with strong positive selection, shaped the evolution of two new ampliconic protein-coding gene families. We propose that, in addition to positive selection, sexual antagonism, and genetic conflict, differences in the usage of DNA repair pathways may give rise to the unique patterns of Y-linked mutations. Together these effects may drive the convergent evolution of Y chromosome structure across taxa.

## Results

### Improving Y chromosome assemblies using long-read assembly and fluorescence in situ hybridization (FISH)

Long reads have enabled the assembly of many repetitive genome regions but have had limited success in assembling Y chromosomes (*Bachtrog et al., 2019*; *Peichel et al., 2019*; *Hall et al., 2016*; *Chang and Larracuente, 2019*). To improve Y chromosome assemblies for comparative genomic analyses, we applied our heterochromatin-sensitive assembly pipeline (*Chang and Larracuente, 2019*) with long reads that we previously generated (*Chakraborty et al., 2021*) to de novo reassemble the Y chromosome from the three species in the *Drosophila simulans* clade. We also resequenced male genomes using PCR-free Illumina libraries to polish these assemblies. Our heterochromatin-enriched methods improve contiguity compared to previous *D. simulans* clade assemblies. We recovered all known exons of the 11 canonical Y-linked genes conserved across the *melanogaster* group, including 58 exons missed in previous assemblies (*Supplementary file 1*; *Gepner and Hays, 1993*; *Bernardo Carvalho et al., 2009*). Based on the median male-to-female coverage (*Chang and Larracuente, 2019*), we assigned 13.7–18.9 Mb of Y-linked sequences per species with N50 ranging from 0.6 to 1.2 Mb. The quality of these new *D. simulans* clade Y assemblies is comparable to *D. melanogaster* (*Table 1*; *Chang and Larracuente, 2019*). We evaluated our methods by comparing our assignments for every 10 kb window of assembled sequence to its known chromosomal location. Our assignments have 96, 98, and 99% sensitivity and 5, 0, and 3% false-positive rates in *D. mauritiana*, *D. simulans*, and *D. sechellia*, respectively (*Supplementary file 2*). We have lower confidence in our *D. mauritiana* assignments, because the male and female Illumina reads are from different library construction methods. Therefore, we applied an additional criterion only in *D. mauritiana* based on the female-to-male total mapped reads ratio ( < 0.1), which reduces the false-positive rate from 13% to 5% in regions with known chromosomal location (*Supplementary file 2*; *Figure 1—figure supplement 1*). We can detect potential misassemblies by looking for discordant assignments between 10 kb windows on the same contigs. Because we do not find any obviously discordant F/M ratios for any contigs, we make chromosome assignments based on median male-to-female coverage and the ratio of female-to-male total mapped reads across whole contigs. Based on these chromosome assignments, we find 40–44% lower PacBio coverage on Y than X chromosomes in all three species (*Figure 1—figure supplement 2*; see Appendix 1).

The cytological organization of the *D. simulans* clade Y chromosomes is not well-described (*Lemeunier and Ashburner, 1984*; *Roy et al., 2005*; *Berloco et al., 2005*). Therefore, we generated new physical maps of the Y chromosomes by combining our assemblies with cytological data. We performed FISH on mitotic chromosomes using probes for 12 Y-linked sequences (*Figure 1* and *Figure 1—figure supplements 3–4*; *Supplementary file 3*) to determine Y chromosome organization at the cytological level. We also determined the location of the centromeres using immunostaining with a Cenp-C antibody (*Figure 1—figure supplement 4*; *Erhardt et al., 2008*). These cytological

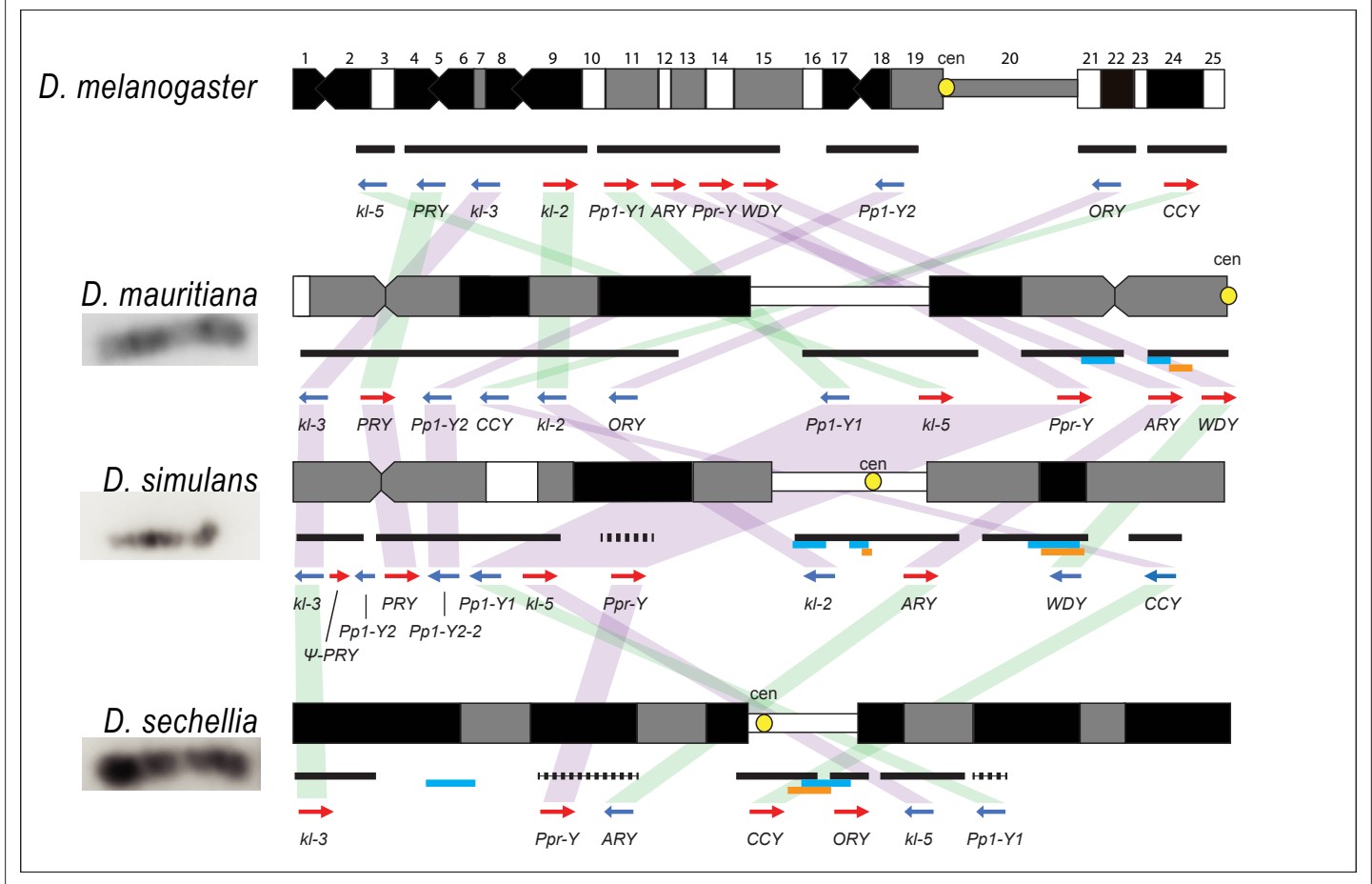

**Figure 1.** Y chromosome organization in *D. melanogaster* and the three *D. simulans* clade species. Schematics of the cytogenetic maps note the locations of Y-linked genes in *D. melanogaster* and *D. simulans* clade species. The bars show the relative placement of the scaffolds on the cytological bands based on FISH results. The solid black and dotted bars represent the scaffolds with known and unknown orientation information, respectively. The light blue and orange bars represent two new Y-linked gene families, *Lhk* and *CK2ßtes-Y* in the *D. simulans* clade, respectively. The arrows indicate the orientation of the genes (blue- minus strand; red- plus strand). Yellow circles denote centromere locations (cen). The blocks connecting genes between species highlight the structural rearrangements between species (purple for same, and green for inverted, orientation).

The online version of this article includes the following figure supplement(s) for figure 1:

**Figure supplement 1.** The distribution of female-to-male total mapped read ratio in each 10-kb window in *D. mauritiana*.

**Figure supplement 2.** The low Pacbio coverage on Y chromosomes in the *D. simulans* clade.

**Figure supplement 3.** Summarized cytological location of satellite DNA, gene families, and conserved genes on the Y chromosome of the *D. simulans* clade.

**Figure supplement 4.** FISH for satellite and gene families, and conserved genes in the *D. simulans* clade.

**Figure supplement 5.** The length of rDNA elements across chromosomes in *D. melanogaster* and the *D. simulans* clade.

data permit us to (1) validate our assemblies and (2) infer the overall organization of the Y chromosome by orienting our scaffolds on cytological maps. Of the 11 Y-linked genes, we successfully ordered 10, 11, and 7 genes on the cytological bands of *D. simulans*, *D. mauritiana*, and *D. sechellia*, respectively (*Figure 1* and *Figure 1—figure supplement 3*). Although we cannot examine the detailed organization as a complete contiguous Y-linked sequence, we did not find any discordance between our scaffolds and cytological data. We find evidence for extensive Y chromosomal structural rearrangements, including changes in satellite distribution, gene order, and centromere position. These rearrangements are dramatic even among the *D. simulans* clade species, which diverged less than 250 KYA (*Figure 1* and *Figure 1—figure supplement 3*). The Y chromosome centromere position appears to be the same as determined by Berloco et al. for different strains of *D. simulans* and

*D. mauritiana*, but not for *D. sechellia* (*Berloco et al., 2005*). One explanation for this discrepancy could be between-strain variation in *D. sechellia* Y chromosome centromere location. Together, our new physical maps and assemblies provide both large and fine-scale resolution on Y chromosome organization in the *D. simulans* clade.

## Y-linked sequence and copy number divergence across three species

Although the *D. simulans* clade species diverged only recently, Y chromosome introgression between pairs of species disrupts male fertility and influences patterns of genome-wide gene expression (*Araripe et al., 2016*; *Johnson et al., 1992*). One candidate locus that may contribute to functional divergence and possibly hybrid lethality is the Y-linked rDNA (*Zhou et al., 2012*; *Paredes et al., 2011*). Y-linked rDNA, specifically 28 S rDNA, were lost in *D. simulans* and *D. sechellia*, but not in *D. mauritiana* (*Roy et al., 2005*; *Lohe and Roberts, 2000*; *Lohe and Roberts, 1990*). However, the intergenic spacer (IGS) repeats between rDNA genes, which are responsible for X-Y pairing in *D. melanogaster* males (*McKee and Karpen, 1990*), are retained on both sex chromosomes in all three species (*Roy et al., 2005*; *Lohe and Roberts, 2000*; *Lohe and Roberts, 1990*). Consistent with previous cytological studies (*Roy et al., 2005*; *Lohe and Roberts, 2000*; *Lohe and Roberts, 1990*), we find that *D. simulans* and *D. sechellia* lost most Y-linked 18 S and 28 S rDNA sequences (*Figure 1—figure supplement 5*). Our assemblies indicate that, despite this loss of the rRNA coding sequences, all three species still retain IGS repeats. However, we and others do not detect Y-linked IGS repeats at the cytological level in *D. sechellia* (*Figure 1—figure supplements 3–4*; *Roy et al., 2005*; *Lohe and Roberts, 2000*; *Lohe and Roberts, 1990*), suggesting that their abundance is below the level of detection by FISH in this species.

Structural variation at Y-linked genes may also contribute to functional variation and divergence in the *D. simulans* clade. Previous studies reported many duplications of canonical Y-linked genes in *D. simulans* (*Helleu et al., 2019*; *Chakraborty et al., 2021*; *Kopp et al., 2006*). We find that all three species have at least one intact copy of the 11 canonical Y-linked genes, but there is also extensive copy number variation in Y-linked exons across these species (*Figure 2* and *Figure 2—figure supplements 1–2*; *Supplementary file 1*; *Chakraborty, 2020*). Using Illumina reads, we confirm the copy number variation in our assemblies and reveal some duplicated Y-linked exons, particularly in *kl-3*, *WDY*, and *Ppr-Y*, that are not assembled in *D. sechellia* (*Figure 2—figure supplement 1*). Some duplicates may be functional because they are expressed and have complete open reading frames, (*e.g. ARY*, *Ppr-Y1*, and *Ppr-Y2*). The *D. simulans* Y chromosome has four complete copies of *ARY*, all of which show similar expression levels from RNA-seq data (*Figure 2B* and *Supplementary file 4*), but two copies have inverted exons 1 and 2. *D. sechellia* also contains at least five duplicated copies of *ARY*, some of which also have the inverted exons 1 and 2, but the absence of RNA-seq data from testes of this species prevents inferences regarding whether all copies of *ARY* are expressed. However, most duplications include only a subset of exons, and in many cases, the duplicated exons are located on the periphery of the presumed functional gene copy (*Figure 2B* and *Figure 2—figure supplement 2*, *Supplementary file 4*). For example, both *D. simulans* and *D. mauritiana* have multiple copies of exons 8–12 located at the 3′ end of *kl-2* (*Figure 2B*). In *D. simulans*, most of these extra exons have low to no expression, while in *D. mauritiana*, there appears to be a substantial expression from many of the duplicated terminal exons, as well as an internal duplication of exon 5. Although the duplications of Y-linked genes can influence their expression, especially for genes with short introns (*e.g. ARY*, *Ppr-Y1* and *Ppr-Y2*), it is unclear what effects these duplicated exons have on the protein sequences of these fertility-essential genes.

All exon-intron junctions are conserved within full-length copies of the canonical Y-linked genes, but intron lengths vary between these species (*Figure 3*). The length of longer introns ( > 100 bp in any species) is more dynamic than that of short introns (*Figure 3*; *Supplementary file 5*). The dramatic size differences in most introns cannot be attributed to a single deletion or duplication (see *ORY* example in *Figure 2—figure supplement 3*). Some Y-linked genes contain mega-base sized introns (*i.e.*, mega-introns) whose transcription manifests as cytologically visible lampbrush-like loops (Y-loops) in primary spermatocytes (*Bonaccorsi et al., 1988*; *Bonaccorsi et al., 1990*). While Y-loops are found across the *Drosophila* genus (*Meyer, 1963*; *Piergentili, 2007*), their potential functions are unknown (*Fingerhut et al., 2019*; *Redhouse et al., 2011*; *Pisano et al., 1993*; *Piergentili et al., 2004*; *Piergentili and Mencarelli, 2008*) and the genes/introns that produce Y-loops differs among species

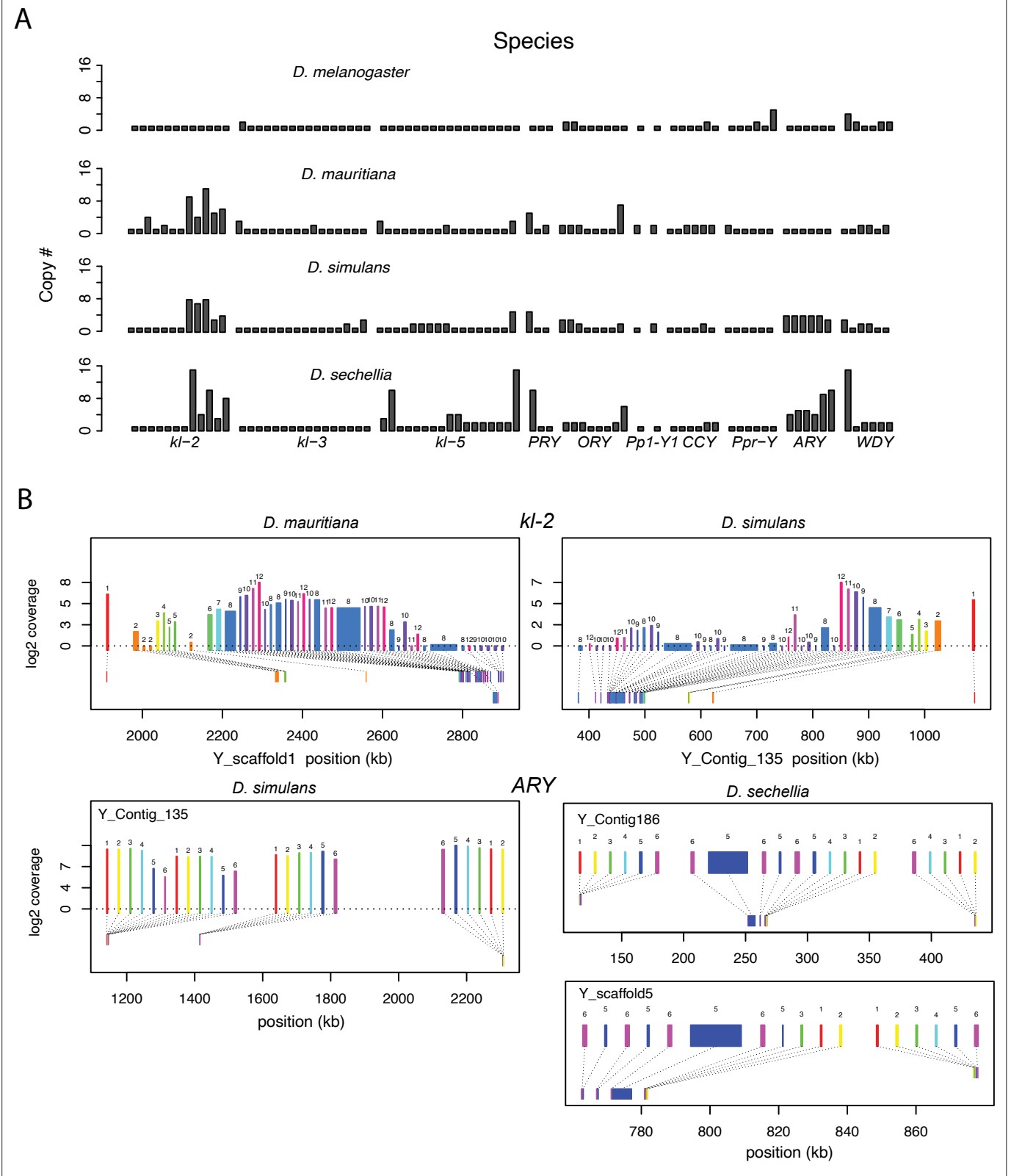

**Figure 2.** Duplication of canonical Y-linked exons. (**A**) Exon copy number is highly variable across the three *D. simulans* clade species and generally greater than in *D. melanogaster*. (**B**) Gene structure of *kl-2* and *ARY* inferred from assemblies and RNA-seq data. Upper bars indicate exons that are colored and numbered, with their height showing average read depth from sequenced testes RNA (*D. simulans* and *D. mauritiana* only). Lower bars indicate exon positions on the assembly and position on the Y-axis indicates coding strand. Some of the duplicated exons are expressed. For short

*Figure 2 continued on next page*

*Figure 2 continued*

genes (*e.g., ARY*), the duplicates may be functional and influence protein expression level, unlike duplicated exons of long genes (*e.g., kl-2*).

The online version of this article includes the following figure supplement(s) for figure 2:

**Figure supplement 1.** The coverage of male Illumina DNA-seq reads in 11 canonical Y-linked genes.

**Figure supplement 2.** Gene structure of 11 conserved Y-linked genes inferred from assemblies and RNA-seq data.

**Figure supplement 3.** The mummerplot of the *ORY* alignment in the *D. simulans* clade.

(*Chang and Larracuente, 2017*). *D. melanogaster* has three Y-loops transcribed from introns of *ORY* (*ks-1* in previous literature), *kl-3*, and *kl-5* (*Bonaccorsi et al., 1988*). Based on cytological evidence, *D. simulans* has three Y-loops, whereas *D. mauritiana* and *D. sechellia* only have two (*Piergentili, 2007*). Of all potential loop-producing introns, we find that only the *kl-3* mega-intron is conserved in all four species and has the same intron structure and sequences (*i.e.* $(AATAT)_n$ repeats). While both *kl-5* and *ORY* produce Y-loops with $(AAGAC)_n$ repeats in *D. melanogaster*, $(AAGAC)_n$ is missing from the genomes of the *D. simulans* clade species. This observation is supported by our assemblies, the Illumina raw reads (*Supplementary file 6*), and published FISH results (*Jagannathan et al., 2017*). In the *D. simulans* clade, the *ORY* introns do not carry any long tandem repeats. However, *kl-5* has introns with $(AATAT)_n$ repeats that may form a Y-loop in the *D. simulans* clade species. These data suggest that, while mega-introns and Y-loops may be conserved features of spermatogenesis in *Drosophila*, they turn over at both the sequence and gene levels over short periods of evolutionary time (*i.e.* ~ 2 My between *D. melanogaster* and the *D. simulans* clade).

Consistent with previous studies (*Tobler et al., 2017*; *Chakraborty et al., 2021*), we identify high rates of gene duplication to the *D. simulans* clade Y chromosome from other chromosomes. We find 49 independent duplications to the Y chromosome in our heterochromatin-enriched assemblies (*Figure 4*; *Supplementary file 7*), including eight newly discovered duplications (*Tobler et al., 2017*; *Chakraborty et al., 2021*). Twenty-eight duplications are DNA-based, 13 are RNA-based, and the rest are unknown due to limited sequence information (*Supplementary file 7*). The rate of transposition to the Y chromosome is about three to four times higher in the *D. simulans* clade compared to *D. melanogaster* (*Chang and Larracuente, 2019*). We also infer that 17 duplicated genes were independently deleted from *D. simulans* clade Y chromosomes. Some of these Y-linked duplications, including *Fdy*, *Mst77Y* and *pirate*, are known to be functional and/or under purifying selection (*Tobler et al., 2017*; *Krsticevic et al., 2015*; *Russell and Kaiser, 1993*; *Chen et al., 2021*). However, based on transcriptomes from *D. simulans* and *D. mauritiana* testes, we suspect that more than half of the duplicated genes are likely pseudogenes that either show no expression in testes ( < 3 TPM) or lack open reading

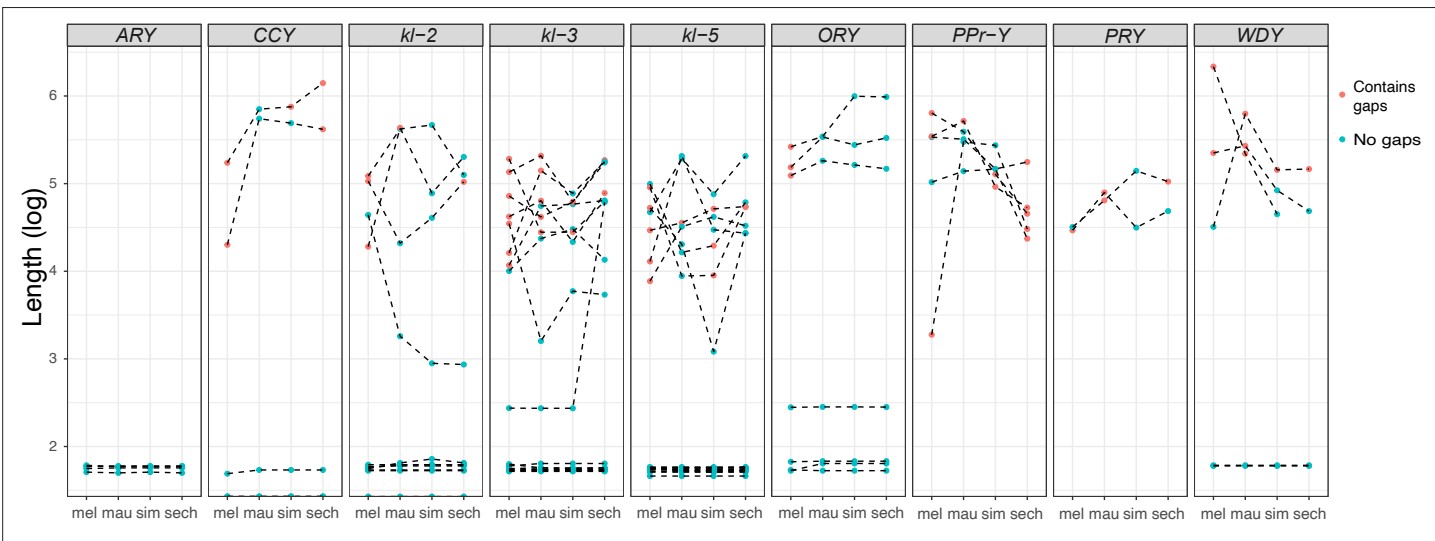

**Figure 3.** Evolution of intron lengths in canonical Y-linked genes. The intron length in canonical Y-linked genes is different between *D. melanogaster* and the three *D.* simulans clade species. Orthologous introns are connected by dotted lines. Completely assembled introns are in blue and introns with gaps in the assembly are in red, and are therefore minimum intron lengths.

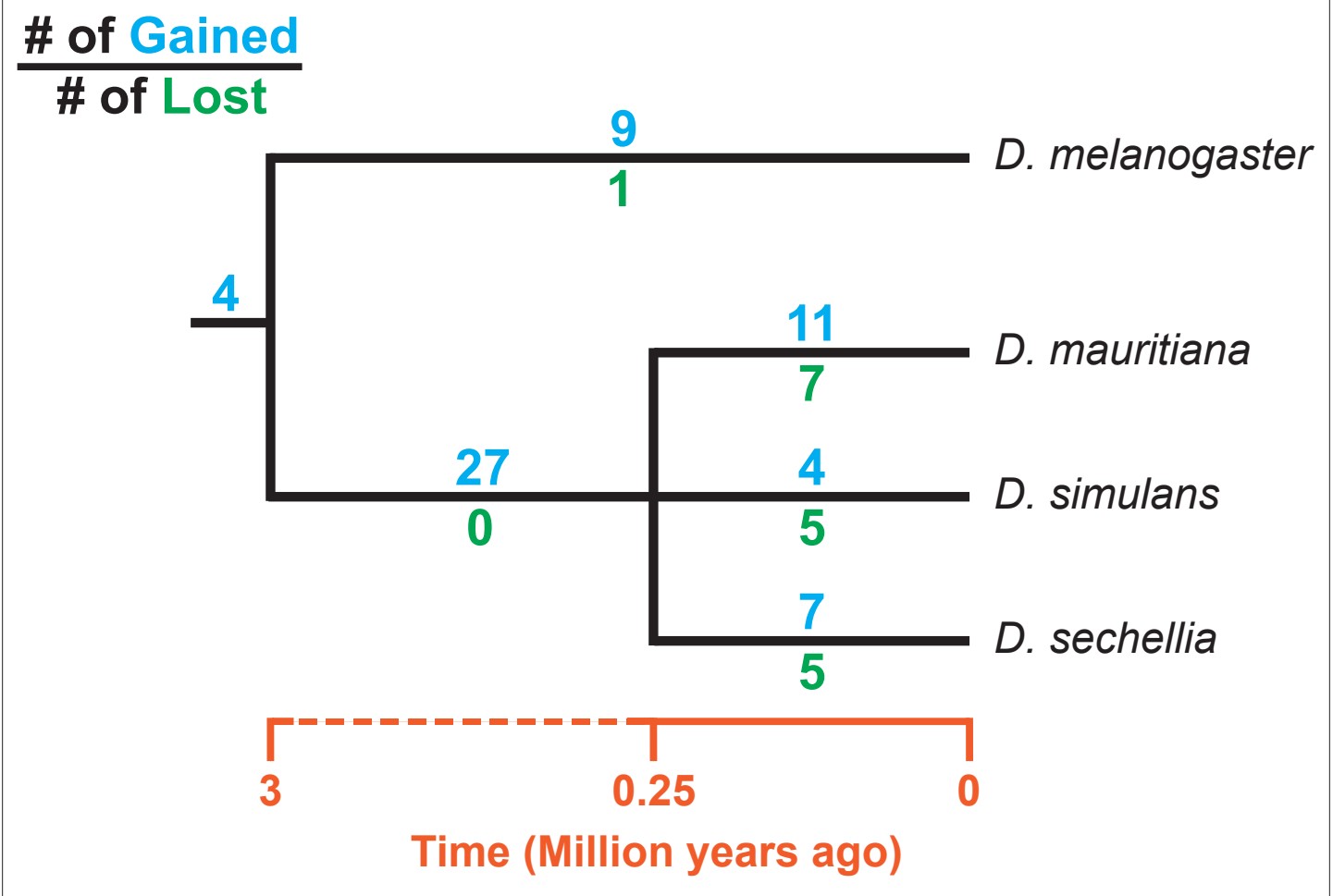

**Figure 4.** Turnover of new duplications to Y chromosomes in *D. melanogaster* and three species in the *D. simulans* clade. Using phylogenetic analyses, we inferred the evolutionary histories of new Y-linked duplications. The blue and green numbers represent the number of independent duplications and deletions observed in each branch, respectively. We also detected four duplications presented in the ancestor of these four species. The deletion events that happened in the ancestor of these four species cannot be inferred without a Y chromosome assembly in the outgroup.

frames ( < 100 amino acids; *Supplementary file 7*). We also detect intrachromosomal duplications of these Y-linked pseudogenes (*Supplementary file 7*), suggesting a high duplication rate within these Y chromosomes.

Most new Y-linked duplications in *D. melanogaster* and the *D. simulans* clade are from genes with presumed functions in chromatin modification, cell division, and sexual reproduction (*Supplementary file 8*), consistent with other *Drosophila* species (*Bachtrog et al., 2019*; *Mahajan and Bachtrog, 2017*). While Y-linked duplicates of genes with these functions may be selectively beneficial, a duplication bias could also contribute to this enrichment, as genes expressed in the testes may be more likely to duplicate to the Y chromosome due to its open chromatin structure and transcriptional activity during spermatogenesis (*Greil and Ahmad, 2012*; *Mahadevaraju et al., 2021*; *Hess and Meyer, 1968*).

## The evolution of new Y-linked gene families

Ampliconic gene families are found on Y chromosomes in multiple *Drosophila* species (*Ellison and Bachtrog, 2019*). We discovered two new gene families that have undergone extensive amplification on *D. simulans* clade Y chromosomes (*Figure 5*). Both families appear to encode functional protein-coding genes with complete open reading frames and high expression in mRNA-seq data (*Supplementary file 9*) and have 36–146 copies in each species' Y chromosome. We also confirm that >90%

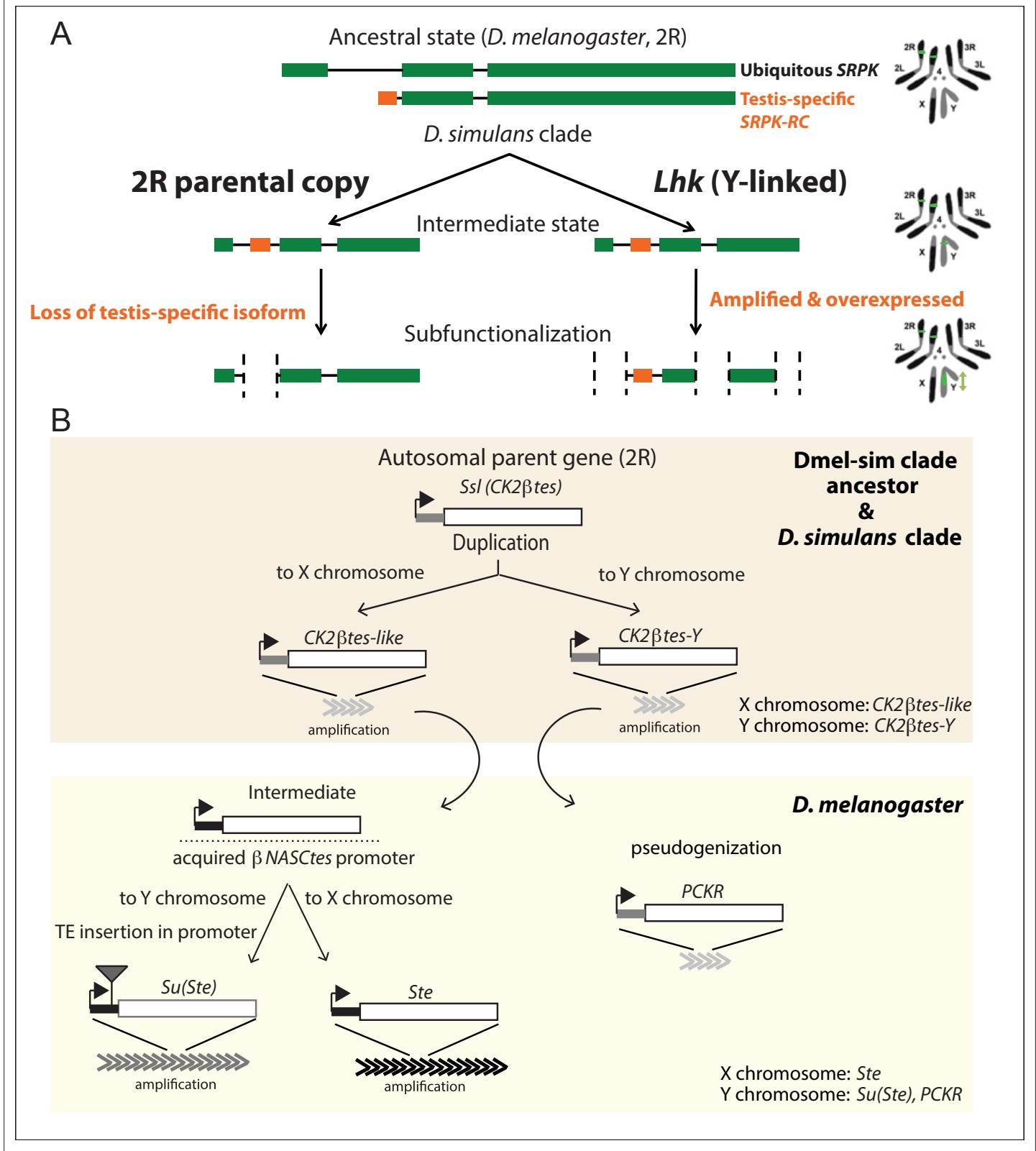

**Figure 5.** The history of Y-linked ampliconic genes. (**A**) Schematic showing the inferred evolutionary history of *SRPK-Y*. *SRPK* duplicated to the ancestral Y chromosome in the *D. simulans* clade. The Y-linked copy (*Lhk*) retained an exon with testis-specific expression, which was lost in the parental copy on 2R. The Y-linked copy (*Lhk*) further duplicated and increased their expression in testes. (**B**) Schematic showing the inferred evolutionary history of sex-linked *Ssl/CK2ßtes* paralogs. In the *D. melanogaster – D. simulans* clade ancestor, the autosomal gene *Ssl/CK2ßtes* duplicated from chromosome

*Figure 5 continued on next page*

*Figure 5 continued*

*2R* to the sex chromosome and independently amplified into the multi-copy gene families *CK2ßtes-like* on the X chromosome and *CK2ßtes-Y* on the Y chromosomes (shaded orange box). The gene structures are maintained in the *D. simulans* clade species, but not in *D. melanogaster*. In the *D. melanogaster* lineage (shaded yellow box), *CK2ßtes-Ys* became pseudogenes (*PCKR*) and *CK2ßtes-like* acquired a promoter from *ßNASCtes* to create a chimeric gene. Subsequent duplication of the chimeric gene to the X chromosome gave rise to the X-linked *Ste* loci in *D. melanogaster*. Duplication of the chimeric gene to the Y chromosome, with a subsequent TE insertion in the promoter and amplification event, gave rise to the Y-linked *Su(Ste)* loci in *D. melanogaster*.

of the variants in our assembled Y-linked gene families are represented in Illumina DNA-seq data (Appendix 1).

The first amplified Y-linked gene family, *SR Protein Kinase* (*SRPK*), is derived from an autosome-to-Y duplication of the sequence encoding the testis-specific isoform of the gene *SR Protein Kinase (SRPK)*. After the duplication of *SRPK* to the Y chromosome, the ancestral autosomal copy subsequently lost its testis-specific exon via a deletion (*Figure 5A*). The movement of the male-specific isoform inspired us to name the Y-linked *SRPK* gene family *Lo-han-kha (Lhk)*, which is the Taiwanese term for the male vagabonds that moved from mainland China to Taiwan during the Qing dynasty. In *D. melanogaster*, *SRPK* is essential for both male and female reproduction (*Loh et al., 2012*). We therefore hypothesize that the relocation of the testis-specific isoform to the *D. simulans* clade Y chromosomes may have resolved intralocus sexual antagonism over these two functions. Our phylogenetic analysis identified two subfamilies of *Lhk* that we designate *Lhk-1* and *Lhk-2* (*Figure 6A*). Both subfamilies are shared by all *D. simulans* clade species and show a 5.5% protein divergence between species. The two subfamilies are found in different locations in our Y chromosome assemblies; consistent with this observation, we detect two to three *Lhk* foci on Y chromosomes in the *D. simulans* clade using FISH (*Figure 6A and C* and *Figure 1—figure supplements 3–4*).

The second amplified gene family comprises both X-linked and Y-linked duplicates of the *Ssl* gene located on chromosome 2 R; it is unclear whether the X- or Y-linked copies originated first (*Figure 5B*). The X-linked copies are known as *CK2ßtes-like* in *D. simulans* (*Kogan et al., 2012*). The Y-linked copies are also found in *D. melanogaster* but are degenerated and have little or no expression (*Chang and Larracuente, 2019*; *Danilevskaya et al., 1991*), leading to their designation as pseudogenes. In the *D. simulans* clade species, however, the Y-linked paralogs have high levels of expression ( > 50 TPM in testes, *Supplementary file 9*) and complete open reading frames, so we refer to this gene family as *CK2ßtes-Y*. Both *CK2ßtes-like* (4–9 copies) and *CK2ßtes-Y* (36–123 copies based on the assemblies) are amplified on the X and Y chromosome in the *D. simulans* clade relative to *D. melanogaster* (*Supplementary file 9*; *Kogan et al., 2012*). The Y-linked copies in *D. melanogaster, Su(Ste),* are known to be a source of piRNAs (*Aravin et al., 2004*). We did not detect any testis piRNAs from either gene family in two small RNA-seq datasets (SRR7410589 and SRR7410590); however, we do find some short ( < 23 nt) reads (0.003–0.005% of total mapped reads) mapped to these gene families (*Supplementary file 10*).

We inferred gene conversion rates and the strength of selection on these Y-linked gene families using phylogenetic analyses on coding sequences. We estimated the gene conversion rate in *D. simulans* clade Y-linked gene families based on four-gamete tests and gene similarity (*Rozen et al., 2003*; *Chang and Larracuente, 2019*; *Ohta, 1984*; *Backström et al., 2005*). In general, *D. simulans* clade species show similar gene conversion rates (on the order of $10^{-4}$ to $10^{-6}$) in both of these families compared to our previous estimates in *D. melanogaster* (*Chang and Larracuente, 2019*; *Supplementary file 11*). These higher gene conversion rates compared to the other chromosomes might be a shared feature of Y chromosomes across taxa (*Rozen et al., 2003*).

To estimate rates of molecular evolution, we conducted branch-model and branch-site-model tests on the reconstructed ancestral sequences of *Lhk-1, Lhk-2, CK2ßtes-Y,* and two *CK2ßtes-like* using PAML (*Figure 6A and B*; *Table 2*; *Yang, 1997*). We used reconstructed ancestral sequences for our analyses to avoid sequencing errors in the assemblies, which appear as singletons. We infer that after the divergence of *D. simulans* clade species, *Lhk-1* evolved under purifying selection, whereas *Lhk-2* evolved under positive selection (*Figure 6A*; *Table 2*; *Figure 6—figure supplement 1*; *Supplementary file 12*). Using transcriptome data, we observe that highly expressed *Lhk-1* copies have fewer nonsynonymous mutations than lowly expressed copies in *D. simulans*, consistent with purifying selection (Chi-square test's p = 0.01; *Figure 6—figure supplement 2* and *Supplementary file 13*). Both

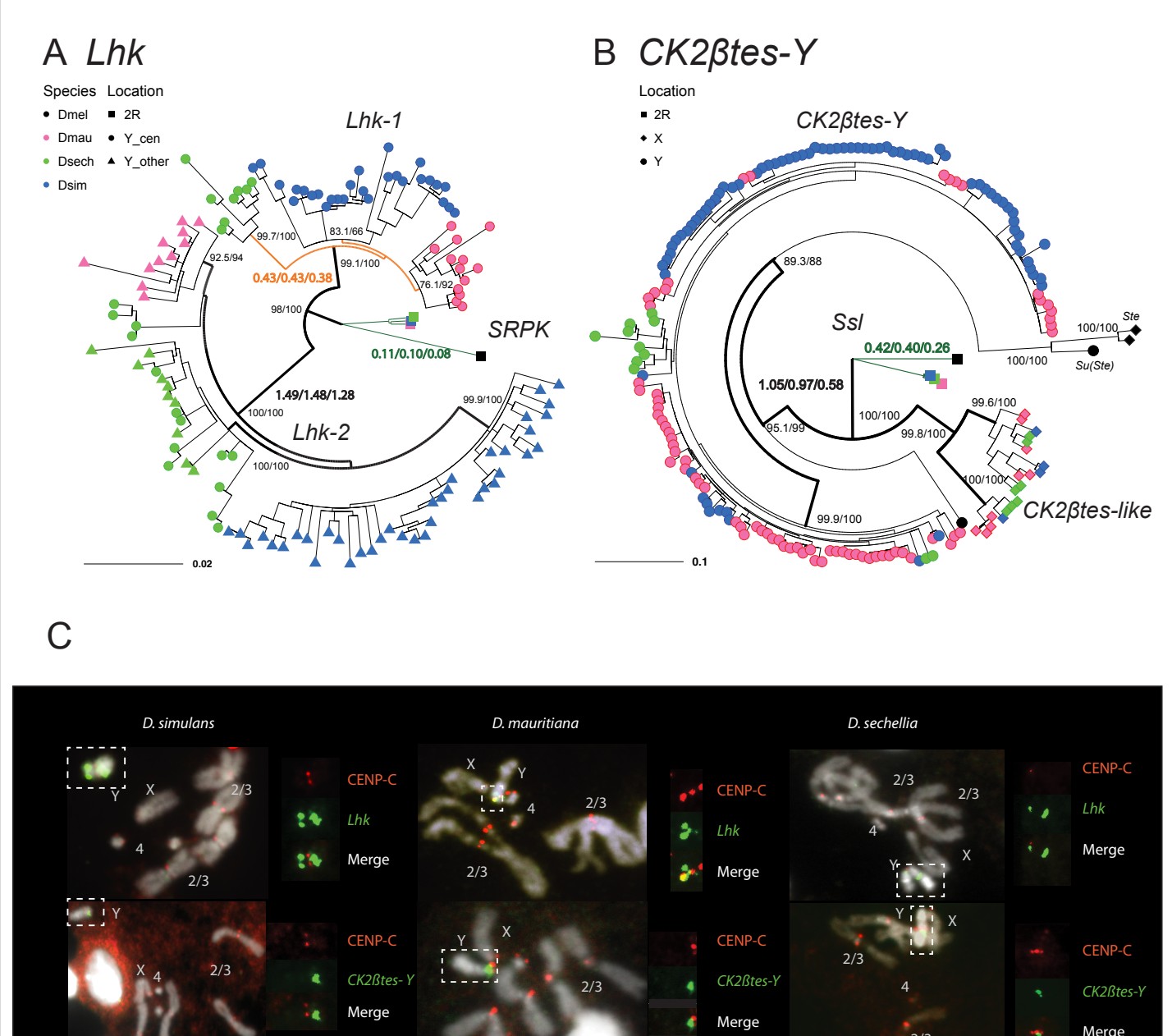

**Figure 6.** The rapid evolution and gene conversion of Y-linked ampliconic genes. (**A**) The inferred maximum likelihood phylogeny for *Lhk*. Node labels indicate SH-aLRT and ultrafast bootstrap (*e.g.* 100/100) or rates of protein evolution from PAML with CodonFreq = 0,1, or 2 (*e.g.* 1.01/1.02/1.03) (*Figure 6—figure supplement 1* and *Figure 6—figure supplement 3*). *Lhk* shows evidence for positive selection (branch tests and branch-site tests with ω >1) after the duplication from 2R (*SRPK*) to the Y chromosome in the *D. simulans* clade. One *Lhk* subfamily (*Lhk-1*) is under recent purifying selection and is located close to the centromere, but the other (*Lhk-2*) is rapidly evolving across the species of the *D. simulans* clade. (**B**) Same as A but for *CK2βtes-Y*. Both Y-linked *CK2βtes-Y* and X-linked *CK2βtes-like* also show positive selection. All ω values shown are statistically significant (LRT tests, P0.05; *Supplementary file 12* and *Supplementary file 14*). (**C**) Cytological location of Y-linked gene families detected using Immunolabeling with fluorescence in situ hybridization (immunoFISH) for the centromere (CENP-C antibody, red signal). On the Y chromosomes, *Lhk* FISH signals suggest that this gene family occurs in 2–3 cytological locations (green signal), with one near the centromere. *CK2βtes-Y* FISH signals are only located near centromeres. Based on our analysis of sequence information, we suggest that most *Lhk-1* copies are located near *CK2βtes-Y* and the centromere.

The online version of this article includes the following figure supplement(s) for figure 6:

*Figure 6 continued on next page*

**Figure supplement 1.** The phylogeny of *Lhk* used in PAML analyses.

**Figure supplement 2.** The expression of different copies from *Lhk* and *CK2ßtes-Y* gene families.

**Figure supplement 3.** The phylogeny of *CK2ßtes-Y* used in PAML analyses.

*Lhk* gene families are expressed two- to seven-fold higher than the ancestral copy on 2R in the same species, and 1.9–64-fold higher than their ortholog, *SRPK-RC,* in *D. melanogaster,* suggesting that gene amplification may confer increased expression. In both *D. simulans* and *D. mauritiana, Lhk-1* is shorter due to deletions following its origin and has a higher expression level than *Lhk-2.* Both *Lhk* gene families have higher copy numbers in *D. simulans* than *D. mauritiana,* which likely contributes to their higher expression level in *D. simulans* (*Supplementary file 9*). For both *Lhk-1* and *Lhk-2,* copies from the same species are more similar than copies from other species—a signal of concerted evolution (*Dover, 1982*).

The ancestral *Ssl* gene experienced a slightly increased rate of protein evolution after it duplicated to the X and Y chromosomes ($\omega$ = 0.41 vs 0.23; p = 0.03; *Figure 6B*; *Table 2*; *Figure 6—figure supplement 3*; *Supplementary file 14*). We find that both *CK2ßtes-like* and *CK2ßtes-Y* share strong signals of positive selection, based on branch-model and branch-site-model tests (p = 8.8E-9; *Figure 6B*; *Table 2*; *Figure 6—figure supplement 3*; *Supplementary file 14*). In *D. melanogaster,* the over-expression of the *CK2ßtes-like* X-linked homolog, *Stellate,* can drive in the male germline by killing Y-bearing sperm and generating female-biased offspring (*Malone et al., 2015*; *Palumbo et al., 1994*; *Meyer et al., 2004*). We suspect that *CK2ßtes-like* and *CK2ßtes-Y* might have similar functions and may also have a history of conflict. Therefore, the co-amplification of sex-linked genes and positive selection on their coding sequences may be a consequence of an arms race between sex chromosome drivers.

## Y chromosome evolution driven by specific mutation patterns

The specific DNA-repair mechanisms used on Y chromosomes might contribute to their high rates of intrachromosomal duplication and structural rearrangements. Because Y chromosomes lack a homolog, they must repair double-strand breaks (DSBs) by non-homologous end joining (NHEJ) or microhomology-mediated end joining (MMEJ), which relies on short homology (usually > 2 bp) to repair DSBs (*Chan et al., 2010*). Compared to NHEJ, MMEJ is more error-prone and can result in translocations and duplications (*McVey and Lee, 2008*). Preferential use of MMEJ instead of NHEJ could contribute to the high duplication rate and extensive genome rearrangements that we observe on Y chromosomes. To infer the mechanisms of DSB repair on Y chromosomes, we counted indels between Y-linked duplicates and their parent genes for a set of 21 putative pseudogenes. Both NHEJ and MMEJ can generate indels, but NHEJ usually produces smaller indels (1–3 bp) compared to MMEJ ( > 3 bp) (*McVey and Lee, 2008*; *Chang et al., 2017*). We also cataloged short stretches of homology between each duplicate and its parent. To compare Y-linked patterns of DSB repair to other regions of the genome, we measured the size of polymorphic indels in intergenic regions and pseudogenes on the autosomes and X chromosomes from population data in *D. melanogaster* (DGRP; *Huang et al., 2014*) and *D. simulans* (*Signor et al., 2018*). To the extent that these indels do not experience selection, their sizes should reflect the mutation patterns on each chromosome. We observe proportionally more large deletions on Y chromosomes (25.1% of Y-linked indels are ≥10 bp deletions; *Supplementary file 15*) compared to other chromosomes in both *D. melanogaster* (12.8% and 15.2% of indels are ≥10 bp deletions in intergenic regions and pseudogenes) and *D. simulans* (7.3% of indels are ≥10 bp deletions in intergenic regions; all pairwise chi-square's p< 1e-6; *Figure 4A*; *Supplementary file 15*). The pattern of excess large deletions is shared in the three *D. simulans* clade species Y chromosomes but is not obvious in *D. melanogaster* (*Figure 7B*). However, because most (36/41) *D. melanogaster* Y-linked indels in our analyses are from copies of a single pseudogene (*CR43975*), it is difficult to compare to the larger samples in the *simulans* clade species (duplicates from 17 genes). The differences in deletion sizes between the Y and other chromosomes are unlikely to be driven by heterochromatin or the lack of recombination. The non-recombining and heterochromatic dot chromosome has a deletion size profile more similar to the other autosomes in *D. simulans* (10.9% of indels are ≥10 bp deletions), consistent with a previous study using TE sequences

**Table 2.** PAML analyses reveal positive selection on Y-linked campliconic gene families.

| Lhk | Branch test with CodonFreq = 0 | | | | | | | Branch-site test site class | | | | | | |
|---|---|---|---|---|---|---|---|---|---|---|---|---|---|---|
| | ω1 | ω2 | ω3 | L | 2ΔlnL | LRT's P | | ω0 | ω1 | ω2a | ω2b | 2ΔlnL | LRT's P | Positively selected sites (BEB > 0.95)* |
| one ω | 0.17 | | | −3250.74 | | | | | | | | | | |
| two ω† | 0.11 | 1.05 | | −3218.26 | 64.94 | 7.71E-16 | | 0.01 | 1 | 4.87 | 4.87 | 13.04 | 3.05E-04 | I4, H11, V32, V75, N99, Y100, D193, D199 |
| three ω‡ | 0.11 | 1.49 | 0.43 | −3216.30 | 3.92 | 0.05 | | | | | | | | |
| *CK2βtes* | | | | | | | | | | | | | | |
| one ω | 0.35 | | | −3295.01 | | | | | | | | | | |
| two ω§ | 0.25 | 1.05 | | −3272.00 | 46.01 | 1.18E-11 | | 0.05 | 1 | 2.21 | 2.21 | 6.54 | 1.06E-02 | D33, T38, K44, K100, F101, K104, M152, M155 |
| three ω‡ | 0.20 | 0.42 | 1.05 | −3266.33 | 11.35 | 7.56E-04 | | | | | | | | |

*See **Supplementary files 12 and 14** for all sites.

†Autosomal and Y lineage have protein evolution of ω1 and ω2, respectively.

‡See **Supplementary files 12 and 14**, Figure 6—figure supplement 1 and Figure 6—figure supplement 3 for the assignment of lineages.

§Autosomal and sex chromosomal (X and Y) have protein evolution of ω1 and ω2, respectively.

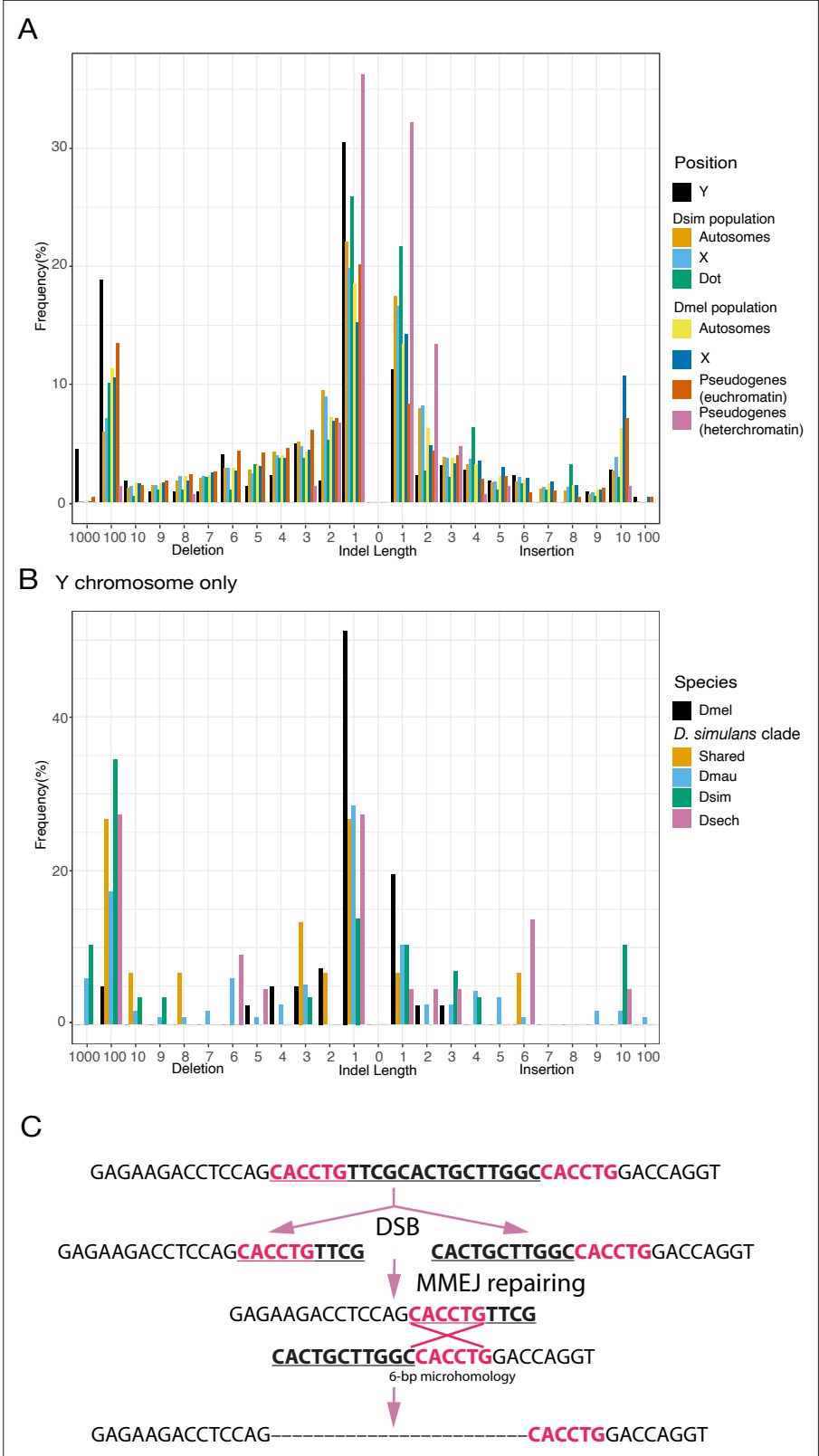

**Figure 7.** An excess of large deletions on Y chromosomes compared to population data suggests a preference for MMEJ. (**A**) We compared the size of 223 indels on 21 recently duplicated Y-linked genes in *D. melanogaster* and the *D. simulans* clade species to the indels polymorphic in the *D. melanogaster* and *D. simulans* populations. For the indels in *D. melanogaster* and *D. simulans* populations, we separated them based on their location,

*Figure 7 continued on next page*

*Figure 7 continued*

including autosomes (excluding dot chromosomes), X chromosomes, and dot chromosomes. We excluded the *D. melanogaster* dot-linked indels due to the small sample size (12). We also surveyed indel polymorphism in pseudogenes in *D. melanogaster* using population data. (**B**) We classify Y-linked indels by whether they are shared between species or specific in one species (**C**) The excess of large deletions (underlined) on the Y chromosomes is consistent with MMEJ between short regions of microhomology (red).

The online version of this article includes the following figure supplement(s) for figure 7:

**Figure supplement 1.** The abundance of repetitive elements on Y chromosomes of *D. melanogaster* and the *D. simulans* clade species.

**Figure supplement 2.** The correlation of TE abundance between Y chromosomes and other chromosomes of *D. melanogaster* and the *D. simulans* clade.

**Figure supplement 3.** The length of LTR retrotransposons between Y chromosomes and other chromosomes of *D. melanogaster* and the *D. simulans* clade.

across different chromatin domains (*Blumenstiel et al., 2002*). We also found fewer large deletions (2/149 indels are ≥10 bp in 400 kb alignments; *Figure 7A*) in heterochromatic pseudogenes using 19 long-read (Pacbio or nanopore) assemblies. The enrichment of 1 bp indels (101/149; *Figure 7A*) in heterochromatic pseudogenes might represent sequencing errors in long-read assemblies (*Weirather et al., 2017*). These results suggest that Y chromosomes may use MMEJ over NHEJ compared to other chromosomes, particularly in the simulans clade species. We also find that across the genome, larger deletions ( > 7 bp) share a similar length of microhomologies for repairing DSBs (39.5–57% deletions have ≥2 bp microhomology; Chi-square test for microhomology length between Y and other chromosomes, p > 0.24; *Supplementary files 15–16*), consistent with most being a consequence of MMEJ-mediated repair.

The satellite sequence composition of Y chromosomes differs between species (*Jagannathan et al., 2017*; *Wei et al., 2018*; *Cechova et al., 2019*). A high duplication rate may accelerate the birth and turnover of Y-linked satellite sequences. We discovered five new Y-linked satellites in our assemblies and validated their location using FISH (*Figure 1—figure supplements 3–4* and *Supplementary file 6*). These satellites only span a few kilobases of sequences (5,515–26,119 bp) and are homogenized. According to its flanking sequence, one new satellite, $(AAACAT)_n$, originated from a DM412B transposable element, which has three tandem copies of AAACAT in its long terminal repeats. The AAACAT repeats expanded to 764 copies on the Y chromosome specifically in *D. mauritiana*. This is consistent with other reports of novel satellites arising from TEs (*Dias et al., 2014*). The other four novel satellites are flanked by transposons ( < 50 bp) and may derive from non-repetitive sequences. The MMEJ pathway may contribute to the birth of new repeats, as this mechanism is known to generate tandem duplications via template-switching during repair (*McVey and Lee, 2008*). Short-tandem repeats can be further amplified via saltatory replication or unequal crossing-over between sister chromatids.

Consistent with findings in other species (*Peichel et al., 2019*; *Chang and Larracuente, 2019*), we find an enrichment of LTR retrotransposons on the *D. simulans* clade Y chromosomes relative to the rest of the genome (*Supplementary file 17*). Interestingly, we find that the Y-linked LTR retrotransposons also turn over between species (*Figure 7—figure supplement 1* and *Supplementary file 18*). We find a positive correlation between the difference in Y-linked TE abundance between *D. melanogaster* and each of the *D. simulans* clade species versus the rest of the genome (rho = 0.45–0.50; *Figure 7—figure supplement 2* and *Supplementary file 18*). This suggests that global changes in transposon activity could explain the differences in Y-linked TEs abundance between species. However, the correlations between species within the *D. simulans* clade are weaker (rho < 0.23; *Figure 7—figure supplement 2* and *Supplementary file 18*), consistent with the possibility that some TEs may shift their insertion preference between chromosomes. To test this hypothesis, we estimated the ages of LTR retrotransposons by their length. We find that the recent insertions of LTR transposons are differently distributed across chromosomes between species (*Figure 7—figure supplement 3*), suggesting that insertion preferences towards genomic regions may differ for some TEs. For example, we detect many recent DIVER element insertions on the Y chromosome in *D. simulans*, but not in *D. sechellia* (*Figure 7—figure supplement 3*).

# Discussion

Despite their independent origins, the degenerated Y chromosomes of mammals, fish, and insects have convergently evolved structural features of gene acquisition and amplification, accumulation of repetitive sequences, and gene conversion. Here, we consider the mutational processes that contribute to this structure and its consequences for Y chromosome biology. Our assemblies revealed extensive Y chromosome rearrangements between three very closely related *Drosophila* species (*Figure 1*). These rearrangements may be the consequence of rejoining telomeres after DSBs, as telomere-specific sequences are embedded in non-telomeric regions of *Drosophila* Y chromosomes (*Berloco et al., 2005*; *Abad et al., 2004*; *Agudo et al., 1999*). We propose that four pieces of evidence suggest DSBs on Y chromosomes may be preferentially repaired using the MMEJ pathway. First, Y-linked sequences are generally absent from the X chromosome, precluding repair of DSBs by homologous recombination in meiosis. Second, NHEJ on Y chromosomes may be limited because the Ku complex, which is required for NHEJ (*Chang et al., 2017*), is excluded from HP1a-rich regions of chromosomes (*Chiolo et al., 2011*). The Ku complex also binds telomeres and might prevent telomere fusions (*Melnikova et al., 2005*; *Samper et al., 2000*), suggesting that a low concentration of Ku on Y chromosomes could also cause high rates of telomere rejoining. Third, the highly repetitive nature of Y chromosomes may increase the rate of DSB formation, which may also contribute to a higher rate of MMEJ (*McVey and Lee, 2008*; *Katsura et al., 2007*). Fourth, we show that Y chromosomes have high duplication and gene conversion rates, and larger deletion sizes than other genomic regions (*Figure 7*), consistent with a preference for MMEJ to repair Y-linked DSBs (*McVey and Lee, 2008*).

The exclusion of the Ku complex from heterochromatin could also contribute to an excess of Y-linked duplications we observe in the *D. simulans* clade relative to *D. melanogaster* (*Figures 2A and 7*). *D. simulans* clade Y chromosomes might harbor relatively more heterochromatin than the *D. melanogaster* Y due to the partial loss of their euchromatic rDNA repeats (*Roy et al., 2005*; *Lohe and Roberts, 2000*; *Lohe and Roberts, 1990*), and *D. simulans* also expresses more heterochromatin-modifying factors, such as *Su(var)*s and *E(var)*s (*Lee and Karpen, 2017*), compared to *D. melanogaster*. To explore these hypotheses, the distribution of the Ku complex across chromosomes in the testes of these species should be studied.

If MMEJ is preferentially used to fix DSBs on the Y chromosome, we might expect that the mutations in the MMEJ pathway would disproportionately impact Y-bearing sperm. Consistent with this prediction, a previous study showed that male *D. melanogaster* with a deficient MMEJ pathway (*DNApol theta* mutants) sire female-biased offspring (*McKee et al., 2000*). Moreover, sperm without sex chromosomes that result from X-Y non-disjunction events are not as strongly affected by an MMEJ deficiency as Y-bearing sperm (*McKee et al., 2000*), suggesting that sperm with Y chromosomes are more sensitive to defects in MMEJ.

*Drosophila* Y chromosomes can act as heterochromatin sinks, sequestering heterochromatin marks from pericentromeric regions and suppressing position-effect variegation (*Brown and Bachtrog, 2017*; *Dimitri and Pisano, 1989*; *Henikoff, 1996*; *Gatti and Pimpinelli, 1992*). Therefore, retrotransposons located in heterochromatin might have higher activities in males due to the presence of Y-linked heterochromatin (*Brown and Bachtrog, 2017*; *Henikoff, 1996*), although the genomic distribution of heterochromatin during spermatogenesis is unknown. We find that, like *D. melanogaster* (*Chang and Larracuente, 2019*), *D. simulans* clade Y chromosomes are enriched in retrotransposons relative to the rest of the genome; however, Y chromosomes from even the closely related *D. simulans* clade species harbor distinct retrotransposons (*Figure 7—figure supplement 1* and *Supplementary file 18*), indicating that some TEs may have rapidly shifted their insertion preference. This preference might benefit the TEs because Y-linked TEs might be expressed during spermatogenesis (*Lawlor et al., 2021*). On the other hand, Y chromosomes can be a significant source of small RNAs that silence repetitive elements during spermatogenesis—for example, *Su(Ste)* piRNAs in *D. melanogaster* (*Quénerch'du et al., 2016*; *Aravin et al., 2001*) —and thus may also contribute to TE suppression. If Y chromosomes contribute to piRNA or siRNA production (*e.g.* have piRNA clusters *Chen et al., 2021*; *Aravin et al., 2001*), then the TE insertion preference for the Y chromosome may sometimes be beneficial for the host, as they could provide immunity against active TEs in males. In this sense, Y chromosomes may even act as "TE traps" that incidentally suppress TE activity in the male germline by producing small RNAs.

Genes may adapt to the Y chromosome after residing there for millions of years (*Wakimoto and Hearn, 1990*; *Hearn et al., 1991*). While most genes that move to the Y chromosome quickly degenerate (*Tobler et al., 2017*; *Carvalho et al., 2015*), a subset of new Y-linked genes are retained, presumably due to important roles in male fertility or sex chromosome meiotic drive. New Y-linked genes may adapt to this unique genomic environment, evolving structures and regulatory mechanisms that enable optimal expression on the heterochromatic and non-recombining Y chromosome (*Dupim et al., 2018*). We identified many Y-linked duplicates in the ~15 Mb of Y chromosome that we surveyed in each species. Future improvements in genomic sequence data and assemblies may recover additional Y-linked duplicates among the unassembled satellite-rich sequences. Here, we describe two new Y-linked ampliconic genes specific to the *D. simulans* clade—*Lhk* and *CK2ßtes-Y*– that show evidence of strong positive evolution and concerted evolution, suggesting that high copy numbers and Y-Y gene conversion are often important for the adaptation of new Y-linked genes.

Many ampliconic genes are taxonomically restricted and are not maintained at high copy numbers over long periods of evolutionary time (*Soh et al., 2014*; *Bachtrog et al., 2019*; *Brashear et al., 2018*; *Ellison and Bachtrog, 2019*; *Hughes et al., 2010*; *Mueller et al., 2008*). Some ampliconic gene families are found on both the X and Y chromosomes (*Ellison and Bachtrog, 2019*; *Malone et al., 2015*; *Cocquet et al., 2012*; *Kruger et al., 2019*; *Lahn and Page, 2000*). While we do not know the function of most such co-amplified gene families, the murine example of *Slx/Slxl1* and *Sly* appears to be engaged in an ongoing arms race between the sex chromosomes (*Cocquet et al., 2012*). We propose that Y-linked gene amplification in the *D. simulans* clade initially occurred due to an arms race and was preserved by gene conversion.

It is intriguing that the *CK2ßtes-like/CK2ßtes-Y* gene family is homologous to the *Ste/Su(Ste)* system in *D. melanogaster* (*Kogan et al., 2012*), which is also hypothesized to play a role in sex-chromosome meiotic drive (*Hurst, 1992*). We speculate that in both the *D. melanogaster* and *D. simulans* clade lineages these gene amplifications have been driven by conflict between the sex chromosomes over transmission through meiosis, but that the conflict involves different molecular mechanisms. In the *CK2ßtes-like/CK2ßtes-Y* system, both X and Y-linked genes are protein-coding genes, which is reminiscent of *Slx/Slxl1* and *Sly* which compete for access to the nucleus where they regulate sex-linked gene expression (*Cocquet et al., 2012*; *Kruger et al., 2019*). In contrast, the Y-linked *Su(Ste)* copies in *D. melanogaster* produce small RNAs that suppress the X-linked *Stellate* (*Aravin et al., 2004*). We propose that *CK2ßtes-like/CK2ßtes-Y* system in the *D. simulans* clade species may represent the ancestral state because the parental gene *Ssl* is a protein-coding gene. We speculate that systems arising from antagonisms between the sex chromosomes may shift from protein-coding to RNA-based over time because, with RNAi, suppression is maintained at a minimal translation cost.

Distinct Y-linked mutation patterns are described in many species (*Soh et al., 2014*; *Rozen et al., 2003*; *Hughes and Page, 2015*; *Bachtrog et al., 2019*; *Tobler et al., 2017*; *Peichel et al., 2019*; *Brashear et al., 2018*; *Hall et al., 2016*). Our analyses provide a link between Y-linked mutation patterns and Y chromosome evolution. While the lack of recombination and male-limited transmission of the Y chromosome reduces the efficacy of selection, the high gene duplication and gene conversion rates may counter these effects and help acquire and maintain new Y-linked genes. The unique Y-linked mutation patterns might be the direct consequence of the heterochromatic environment on sex chromosomes. Therefore, we predict that W chromosomes and non-recombining sex-limited chromosomes (*e.g.* some B chromosomes), may share similar mutation patterns with Y chromosomes. Indeed, W chromosomes of birds have ampliconic genes and are rich in tandem repeats (*Backström et al., 2005*; *Komissarov et al., 2018*). However, there seem to be fewer ampliconic gene families on bird W chromosomes compared to Y chromosomes in other animals, suggesting that sexual selection and intragenomic conflict in spermatogenesis are important contributors to Y-linked gene family evolution (*Bachtrog, 2020*; *Rogers, 2021*).

## Materials and methods
### Assembling Y chromosomes using Pacbio reads in *D. simulans* clade
We applied the heterochromatin-sensitive assembly pipeline from *Chang and Larracuente, 2019*. We first extracted 229,464 reads with 2.2-Gbp in *D. mauritiana*, 269,483 reads with 2.3-Gbp in *D. simulans*, and 257,722 reads with 2.6-Gbp in *D. sechellia* using assemblies from *Chakraborty et al.,*

*2021*, respectively. We then assembled these reads using Canu v1.3 and FALCON v0.5.0 combined the parameter tuning method on two error rates, eM and eg, in bogart to optimize the assemblies. We first made the Canu assemblies using the parameters 'genomeSize = 30 m stopOnReadQuality = false corMinCoverage = 0 corOutCoverage = 100 ovlMerSize = 31' and 'genomeSize = 30 m stopOnRead-Quality = false'. For FALCON v0.5.0, we used the parameters 'length_cutoff = –1; seed_coverage = 30 or 40; genome_size = 30000000; length_cutoff_pr = 1000'. We then picked the assemblies with highest contiguity and completeness without detectable misassemblies from each setting (two Canu settings and one Falcon setting).

After picking the three best assemblies for each species, we tentatively reconciled the assemblies using Quickmerge (*Chakraborty et al., 2016*). We examined and manually curated the merged assemblies. For the *D. mauritiana* assembly, we merged two Canu and one FALCON assemblies, and for our *D. simulans* and *D. sechellia* assemblies, we merged one Canu and one FALCON assemblies independently. We manually curated some conserved Y-linked genes using raw reads and cDNA sequences from NCBI, including *kl-3* of *D. mauritiana*, *kl-3*, *kl-5*, and *PRY* of *D. simulans* and *CCY*, *PRY*, and *Ppr-Y* of *D. sechellia*, due to their low coverage and importance for our phylogenetic analyses. We then merged our heterochromatin restricted assemblies with contigs of the major chromosome arms from *Chakraborty et al., 2021*. We polished the resulting assemblies once with Quiver using PacBio reads (SMRT Analysis v2.3.0; *Chin et al., 2013*) and ten times with Pilon v1.22 (*Walker et al., 2014*) using raw Illumina reads with parameters '--mindepth 3 --minmq 10 --fix bases'.

We identified misassemblies and found parts of Y-linked sequences in the contigs from major arms using our female/male coverage assays in *D. sechellia*. We also assembled the total reads (assuming genome size of 180 Mb) and heterochromatin-extracted reads (assuming genome size 40 Mb) using wtdbg v2.4 with parameters '-x rs -t24 -X 100 -e 2' (*Ruan and Li, 2020*) and Flye v2.4.2 (*Kolmogorov et al., 2019*) with default parameters separately. We polished the resulting wtdbg assemblies with raw Pacbio reads using Flye v2.4.2. We then manually assembled five introns and fixed two misassemblies using sequences from wtdbg whole-genome assemblies (two introns), Flye whole-genome (two introns), and heterochromatin-enriched assemblies (one intron) in *D. sechellia*. We assembled one intron using sequences from wtdbg whole-genome assemblies in *D. simulans*.

We also extracted potential microbial reads (except for *Wolbachia*) that mapped to the *D. sechellia* microbial contigs, and assembled these reads into a 4.5 Mb contig, which represents the whole genome of a *Providencia* species, using Canu v 1.6 (r8426 14,520f819a1e5dd221cc16553cf5b5269227b0a3) with parameters 'genomeSize = 5 m useGrid = false stopOnReadQuality = false corMinCoverage = 0 corOutCoverage = 100'. To detect other symbiont-derived sequences in our assemblies, we used Blast v2.7.1+ (*Altschul et al., 1990*) with blobtools (v1.0; *Laetsch and Blaxter, 2017*) to search the nt database (parameters '-task megablast -max_target_seqs 1 -max_hsps 1 -evalue 1e-25'). We estimated the Illumina coverage of each contig in males for *D. mauritiana*, *D. simulans,* and *D. sechellia*, respectively. We designated and removed contigs homologous to bacteria and fungi in subsequent analyses (*Supplementary file 19*).

## Generating DNA-seq from males in the *D. simulans* clade

We extracted DNA from 30 virgin 0-day males using DNeasy Blood & Tissue Kit and diluted it in 100 μL ddH$_2$O. The DNA was then treated with 1 μL 10 mg/mL RNaseA (Invitrogen) at 37 °C for 1 hr and was re-diluted in 100 μL ddH$_2$O after ethanol precipitation. The size and concentration of DNA were analyzed by gel electrophoresis, Nanodrop, Qubit and Genomic DNA ScreenTape. Finally, we constructed libraries using PCR-free standard Illumina kit and sequenced 125 bp paired-end reads with a 550 bp insert size from the libraries using Hiseq 2500 in UR Genomics Research Center. We deposited the reads in NCBI's SRA under BioProject accession number PRJNA748438.

## Identifying Y-linked contigs

To assign contigs to the Y chromosome, we used Illumina reads from male and female PCR-free genomic libraries (except females of *D. mauritiana*) as described in *Chang and Larracuente, 2019*. In short, we mapped the male and female reads separately using BWA (v0.7.15; *Li and Durbin, 2010*) and called the coverage of uniquely mapped reads per site with samtools (v1.7; -Q 10 *Li et al., 2009*). We further assigned contigs with the median of male-to-female coverage across contigs equal to 0 as Y-linked. We examined the sensitivity and specificity of our methods using all 10 kb regions with

known location. Based on our results for 10 kb regions with known location (*Supplementary file 2*) in *D. mauritiana*, we set up an additional criterion for this species—'the average of female-to-male coverage < 0.1'—to reduce the false discovery rate.

## Gene and repeat annotations

We used the same pipeline and data to annotate genomes as a previous study (*Chakraborty et al., 2021*). We collected transcripts and translated sequences from *D. melanogaster* (r6.14) and transcript sequences from *D. simulans Nouhaud, 2018* using IsoSeq3 (*Gordon et al., 2015*). We mapped these sequences to each assembly to generate annotations using maker2 (v2.31.9; *Holt and Yandell, 2011*). We further mapped the transcriptomes using Star 2.7.3 a 2-pass mapping with the maker2 annotation and parameters '-outFilterMultimapNmax 200 --alignSJoverhangMin 8 --alignSJDBoverhangMin 1 --outFilterMismatchNmax 999 --outFilterMismatchNoverReadLmax 0.04 --alignIntronMin 20 --alignIntronMax 5000000 --alignMatesGapMax 5000000 --outSAMtype BAM SortedByCoordinate --readFilesCommand zcat --peOverlapNbasesMin 12 --peOverlapMMp 0.1'. We then generated the consensus annotations using Stringtie 2.0.3 from all transcriptomes (*Pertea et al., 2015*). We further improved the mitochondria annotation using MITOS2. We assigned predicted transcripts to their homologs in *D. melanogaster* using BLAST v2.7.1+ (-evalue 1e-10; *Altschul et al., 1990*).

We used RepeatMasker v4.0.5 (*Smit et al., 2013*) with our custom library to annotate the assemblies using parameter '-s.' Our custom library is modified from *Chakraborty et al., 2021*, by adding the consensus sequence of *Jockey-3* from *D. melanogaster* to replace its homologs (*G2* in *D. melanogaster* and *Jockey-3* in *D. simulans*; *Chang et al., 2019*). We extracted the sequences and copies of TEs and other repeats using scripts modified from *Bailly-Bechet et al., 2014*. To annotate tandem repeats in assemblies, we used TRFinder (v4.09; *Benson, 1999*) with parameters '2 7 7 80 10 100 2000 -ngs -h'. We also used kseek (*Wei et al., 2018*) to search for tandem repeats in the male Illumina reads.

## Transcriptome analyses

We mapped the testes transcriptome to the reference genomes of *D. melanogaster, D. simulans,* and *D. mauritiana* (*Supplementary file 20*; no available transcriptome from *D. sechellia*). We used Stringtie 2.0.3 (*Pertea et al., 2015*) to estimate the expression level using the annotation. However, we applied a different strategy for estimating expression levels of the Y-linked gene families due to the difficulties in precisely annotating multi-copies genes. We constructed a transcript reference using current gene annotation but replaced all transcripts from *Lhk-1, Lhk-2,* and *CK2ßtes-Y* with their species-specific reconstructed ancestral copies. We then mapped the transcriptome reads to this reference using Bowtie2 v 2.3.5.1 (*Langmead and Salzberg, 2012*) with parameters '-very-sensitive -p 24 k 200 X 1000 --no-discordant --no-mixed'. We then estimated the expression level by salmon v 1.0.0 (*Patro et al., 2017*) with parameters '-l A -p 24.' We also mapped small RNA reads from *D. simulans* testes to our custom repeat library and reconstructed ancestral *Lhk-1, Lhk-2,* and *CK2ßtes-Y* sequences using Bowtie v 1.2.3 (*Langmead, 2010*) with parameters '-v3 -q -a -m 50 --best –strata.'

To assay the specific expression of different copies, we also mapped transcriptomic and male genomic reads to the same reference using BWA (v0.7.15; *Li and Durbin, 2010*). We used ABRA v2.22 (*Mose et al., 2019*) to improve the alignments around the indels of these two gene families. We used samtools (v1.7; *Li et al., 2009*) to pile up reads that mapped to reconstructed ancestral copies and estimated the frequency of derived SNPs in the reads.

## Estimating Y-linked exon copy numbers using Illumina reads

We mapped the Illumina reads from the male individuals of *D. melanogaster* and the *D. simulans* clade species to a genome reference with transcripts of 11 conserved Y-linked genes and the sequences of all non-Y chromosomes (r6.14) in *D. melanogaster*. We called the depth using samtools depth (v1.7; *Li et al., 2009*), and estimated the copy number of each exon using the mapped depth. We assumed most Y-linked exons are single-copy, so we divided the depth of each site by the majority of depth across all Y-linked transcripts to estimate the copy number. For the comparison, we simulated the 50 X Illumina reads from our assemblies using ART 2.5.8 with the parameter (art_illumina -ss HSXt -m 500 s 200 p -l 150 f 50; *Huang et al., 2012*). We then mapped the simulated reads to the same reference, called the depth, and divided the depth of each site by 50.

## Immunostaining and FISH of mitotic chromosomes

We conducted FISH in brain cells following the protocol from *Larracuente and Ferree, 2015* and immunostaining with FISH (immune-FISH) in brain cells following the protocol from *Pimpinelli et al., 2011* and *Chang et al., 2019*. Briefly, we dissected brains from third instar larva in 1 X PBS and treated them for 1 min in hypotonic solution (0.5% sodium citrate). Then, we fixed brain cells in 1.8% paraformaldehyde, 45% acetic acid for 6 min. We subsequently dehydrated in ethanol for the FISH experiments but not for the immune-FISH.

For immunostaining, we rehydrated the slide using PBS with 0.1% TritonX-100 after removing the coverslip using liquid nitrogen. The slides were blocked with 3% BSA and 1% goat serum/ PBS with 0.1% TritonX-100 for 30 min and hybridized with 1:500 anti-Cenp-C antibody (gift from Dr. Barbara Mellone) overnight at 4 °C. We used 1:500 secondary antibodies (Life Technologies Alexa-488, 546, or 647 conjugated, 1:500) in blocking solution with 45 min room temperature incubation to detect the signals. We fixed the slides in 4% paraformaldehyde in 4XSSC for 6 min before doing FISH.

We added probes and denatured the fixed slides at 95 °C for 5 min and then hybridized slides at 30°C overnight. For PCR amplified probes with DIG or biotin labels, we blocked the slides for 1 hr using 3% BSA/PBS with 0.1% Tween and incubated slides with 1:200 secondary antibodies (Roche) in 3% BSA/4 X SSC with 0.1% Tween and BSA at room temperature for 1 hr. We made *Lhk* and *CK2ßtes-Y* probes using PCR Nick Translation kits (Roche) and ordered oligo probes from IDT. We list probe information in *Supplementary file 3*. We mounted slides in Diamond Antifade Mountant with DAPI (Invitrogen) and visualized them on a Leica DM5500 upright fluorescence microscope, imaged with a Hamamatsu Orca R2 CCD camera and analyzed using Leica's LAX software. We interpreted the binding patterns of Y chromosomes using the density of DAPI staining solely.

## Phylogenetic analyses of Y-linked genes

We used BLAST v2.7.1+ (*Altschul et al., 1990*) to extract the sequences of Y-linked duplications and conserved Y-linked genes from the genome. We only used high-quality sequences polished by Pilon (--mindepth 3 --minmq 10) for our phylogenetic analyses. We aligned and manually inspected sequences with reference transcripts from Flybase using Geneious v8.1.6 (*Kearse et al., 2012*). For most Y-linked duplications, except for the genes homologous to *Lhk* and *CK2ßtes-Y*, we constructed neighbor-joining trees using the HKY model with 1000 replicates using Geneious v8.1.6 (*Kearse et al., 2012*) to infer their phylogenies. We also measured the length and microhomology in 223 indels from 21 Y-linked duplications using these alignments (*Supplementary file 15*). We also infer the potential mechanisms causing the indels, including tandem duplications and polymerase slippage during DNA replication. We measured the length and microhomology of polymorphic indels in *D. melanogaster* (DGRP *Huang et al., 2014*) and *D. simulans* (*Signor et al., 2018*) populations from *Chakraborty et al., 2021*. For *Lhk* and *CK2ßtes-Y*, we constructed phylogeny using iqtree 1.6.12 (*Nguyen et al., 2015*; *Hoang et al., 2018*) using parameters "-m MFP -nt AUTO -alrt 1000 -bb 1000 -bnni". The node labels in *Figure 5* correspond to SH-aLRT support (%) / ultrafast bootstrap support (%). The nodes with SH-aLRT ≥ 80% and ultrafast bootstrap support ≥ 95% are strongly supported. Protein evolutionary rates (with CodonFreq = 0/1/2 in PAML) of the bold branches were estimated using PAML with branch models on the reconstructed ancestor sequences (*Figure 6—figure supplement 1* and *Figure 6—figure supplement 3*).

## Estimating recombination and selection on Y-linked ampliconic genes

Using the phylogenetic trees from iqtree, we infer the most probable sequences for the internal nodes using MEGA 10.1.5 (*Kumar et al., 2018*; *Stecher et al., 2020*) using the maximal likelihood method and G + I model with GTR model. We conducted branch and branch-site models tests in PAML 4.8 using the ancestral sequences of Y-linked and X-linked ampliconic gene families with their homologs on autosomes. We plotted the tree using R package ape 5.3 (*Paradis et al., 2004*).

We used compute 0.8.4 (*Thornton, 2003*) to calculate Rmin and population recombination rates based on linkage disequilibrium (*Hudson, 1987*; *Hudson and Kaplan, 1985*) and gene similarity. We included sites with indel polymorphisms in these analyses to increase the sample size (558–1544 bp alignments). We also reanalyzed data from *Chang and Larracuente, 2019* to include variant information from these sites. The high similarity between Y-linked ampliconic gene copies may lead us to

overestimate gene conversion based on gene similarity (*Hudson, 1987*). We therefore also reported the lower bound on the gene conversion rate using Rmin (*Hudson and Kaplan, 1985*).

## GO term analysis

We used PANTHER (Released 20190711; *Mi et al., 2019*) with GO Ontology database (Released 2019-10-08) to perform Biological GO term analysis of new Y-linked duplicated genes using Fisher's exact tests with FDR correction. We input 70 duplicated genes with any known GO terms and used all genes (13,767) in *D. melanogaster* as background.

## Indel analyses

We downloaded the SNP calls (vcf files) from population genomic data in North Carolina of *D. melanogaster* (DGRP *Huang et al., 2014*) and California of *D. simulans* (*Signor et al., 2018*). We then used vcftools (*Danecek et al., 2011*) to remove the low-quality SNPs using parameters '--maf 0.1 --keep-only-indels --min-alleles 2 --max-alleles 2 --recode'. We additionally filtered out the potential mismapped regions with '--max-missing-count 20' in *D. melanogaster* or '--max-missing-count 17' in *D. simulans.* Lastly, we analyzed the SNPs in the specific regions using bedtools intersect (*Quinlan and Hall, 2010*) with gene annotation files (dmel-r5.57 or dsim annotation from maker2 v2.31.9; *Holt and Yandell, 2011*). For the heterochromatic pseudogenes, we download 18 long-read polished assemblies from NCBI (*Supplementary file 20*). We then used blastn to get sequences of pseudogenes from the population, aligned, and surveyed their indel lengths. All the alignments for our indel assignment are available in the GitHub repository (https://github.com/LarracuenteLab/simclade_Y; *Chang, 2022*; copy archived at swh:1:rev:b1939db576cb1616094a59775a38014a7d61eb7f) and the Dryad digital repository (https://doi.org/10.5061/dryad.280gb5mr6).

## Acknowledgements

This work was funded by the National Institutes of Health (NIH) (R35GM119515 to AML and R01GM123194 to CDM), National Science Foundation (NSF MCB 1844693) to AML and funding from the University of Nebraska-Lincoln to CDM. AML was supported by a Stephen Biggar and Elisabeth Asaro fellowship in Data Science. C-HC was supported by the Messersmith Fellowship from the U of Rochester, the Government Scholarship to Study Abroad from Taiwan, and the Damon Runyon fellowship (DRG: 2438–21). We thank our collaborators, Drs. JJ Emerson and Mahul Chakraborty, for generating PacBio reads in the *D. simulans* clade, Dr. Barbara Mellone for the antibodies, and Drs. Casey Bergman, Grace YC Lee, Kevin Wei and Anthony Geneva and Larracuente lab members for helpful discussion. We also thank the U of Rochester CIRC for access to computing cluster resources and UR Genomics Research Center for the library construction and sequencing.

## Additional information

### Funding

| Funder | Grant reference number | Author |
| --- | --- | --- |
| National Institute of General Medical Sciences | R35GM119515 | Amanda M Larracuente |
| National Institute of General Medical Sciences | R01GM123194 | Colin D Meiklejohn |
| National Science Foundation | MCB 1844693 | Amanda M Larracuente |
| Damon Runyon Cancer Research Foundation | DRG: 2438-21 | Ching-Ho Chang |
| College of Arts and Sciences, University of Nebraska-Lincoln | | Colin D Meiklejohn |

| Funder | Grant reference number | Author |
| --- | --- | --- |
| University of Rochester | | Amanda M Larracuente<br>Ching-Ho Chang |
| Ministry of Education, Taiwan | | Ching-Ho Chang |

The funders had no role in study design, data collection and interpretation, or the decision to submit the work for publication.

## Author contributions

Ching-Ho Chang, Conceptualization, Data curation, Formal analysis, Funding acquisition, Investigation, Methodology, Project administration, Supervision, Validation, Visualization, Writing – original draft, Writing – review and editing; Lauren E Gregory, Investigation, Validation, Writing – review and editing; Kathleen E Gordon, Investigation, Writing – review and editing; Colin D Meiklejohn, Data curation, Formal analysis, Funding acquisition, Investigation, Validation, Visualization, Writing – review and editing; Amanda M Larracuente, Conceptualization, Data curation, Formal analysis, Funding acquisition, Investigation, Methodology, Project administration, Resources, Supervision, Validation, Visualization, Writing – original draft, Writing – review and editing

## Author ORCIDs

Ching-Ho Chang http://orcid.org/0000-0001-9361-1190
Colin D Meiklejohn http://orcid.org/0000-0003-2708-8316
Amanda M Larracuente http://orcid.org/0000-0001-5944-5686

## Decision letter and Author response

Decision letter https://doi.org/10.7554/eLife.75795.sa1
Author response https://doi.org/10.7554/eLife.75795.sa2

---

# Additional files

## Supplementary files

• Supplementary file 1. The copy number of exons in conserved Y-linked genes. We listed the copy number of each exon in conserved Y-linked genes based on BLAST results.

• Supplementary file 2. The estimates of sensitivity and specificity of our Y-linked sequence assignment methods using 10 kb regions with known chromosomal location. We calculated the median female-over-male coverage in our Illumina data in every 10 kb region with known chromosomal location. We then estimated the sensitivity and specificity of our methods using these data.

• Supplementary file 3. Probe and primer information.

• Supplementary file 4. The genomic location of duplicated exons in conserved Y-linked genes. We listed the genomic location of each exon in conserved Y-linked genes in our assemblies based on BLAST results.

• Supplementary file 5. The intron length of all conserved Y-linked genes across species. We showed the length of each Y-linked exon in all conserved Y-linked genes based on BLAST results. If there are multiple copies of an exon, we choose the copy with a complete open reading frame and the highest expression level.

• Supplementary file 6. The abundance of simple repeats in Illumina reads from male flies estimated with kseek and from our genome assemblies. We used kseek to measure the relative abundance of simple repeats in our Illumina reads. We also used TRF finder to calculate repeat contents in our assemblies. We compared the two results and picked probes for our FISH experiments.

• Supplementary file 7. Recent Y-linked duplications in *D. melanogaster* and species in the *D. simulans* clade. We list information on the recent Y-linked duplications and genes, including copy numbers, expression levels, phylogenies, and open reading frames. We also included some duplications from repetitive regions where we can date their origins.

• Supplementary file 8. Enriched GO terms in Y-linked duplicated genes in *D. melanogaster* and the *D. simulans* clade. We identified GO terms associated with genes that recently duplicated to the Y chromosome listed in *Supplementary file 7* using PANTHER (Released 20190711; [163]). We listed

all GO terms significantly enriched in the duplication (FDR < 0.05).

• Supplementary file 9. The summary of conserved Y-linked genes and ampliconic genes expression. We summarized the expression level of conserved Y-linked genes and ampliconic genes. We sum up the gene expression for genes with multiple duplicated copies on Y chromosomes.

• Supplementary file 10. The number of small RNA reads mapped to the repetitive sequences and Y-linked gene families in the *D. simulans* clade.

• Supplementary file 11. Gene conversion rates for Y-linked ampliconic genes in the *D. simulans* clade. We listed the gene conversion rates and gene similarities on each Y-linked ampliconic gene family (*e.g.*, *Lhk-1, Lhk-2,* and *CK2ßtes-Y*). We estimated gene conversion rates using both gene similarities (p) and population recombination rates (Rmin and rho).

• Supplementary file 12. PAML results for branch and branch-site model analyses of *Lhk* in the *D. simulans* clade. We showed raw results and LRT tests for branch and branch-site model analyses from PAML. We also report rates of protein evolution for each branch in each model and sites under positive selection in the branch-site model analyses.

• Supplementary file 13. The number of new mutations observed in highly and lowly expressed copies of Y-linked gene families. We list the number of synonymous, nonsynonymous and UTR changes in highly and lowly expressed copies of Y-linked genes families. We suggest that highly expressed copies evolve under stronger selection (positive or purifying) than other copies. Therefore, we compared the number of synonymous changes over nonsynonymous changes in highly expressing copies to the other copies. See *Supplementary file 21* for detailed information.

• Supplementary file 14. PAML results for branch and branch-site model analyses of *CK2ßtes-Y* in the *D. simulans* clade. We showed raw results and LRT tests for branch and branch-site model analyses from PAML. We also report rates of protein evolution for each branch in each model and sites under positive selection in the branch-site model analyses.

• Supplementary file 15. Indels in Y-linked duplications in *D. melanogaster* and the *D. simulans* clade. We listed the position and sizes of all indels we found in Y-linked duplications. We also inferred the potential microhomologies used for MHEJ repairing. We also infer other DSB repairing mechanisms, including tandem duplications and replication slippages, based on the sequence information.

• Supplementary file 16. Polymorphic indels in *D. melanogaster* and *D. simulans* populations. We listed the position and sizes of polymorphic indels from *D. melanogaster* and *D. simulans* populations. We also inferred the potential microhomologies causing the deletions.

• Supplementary file 17. Repeat composition across chromosomes in *D. melanogaster* and the *D. simulans* clade. We list the composition of LTR retrotransposon, LINE, DNA transposons, satellite, simple repeats, rRNA, and other repeats across every chromosome in our assemblies.

• Supplementary file 18. The detail of repetitive sequences across chromosomes in *D. melanogaster* and the *D. simulans* clade. We list the total sequence length from each transposon or complex repeat on Y-linked contigs/scaffolds and other contigs/scaffolds in our assemblies.

• Supplementary file 19. The Illumina coverage and blast result for each contig in the *D. simulans* clade. We used Blast v2.7.1+ [135] with blobtools (v1.0; [136]) to search the nt database (parameters "-task megablast -max_target_seqs 1 -max_hsps 1 -evalue 1e-25"). We estimated the Illumina coverage of each contig in males of *D. mauritiana*, *D. simulans* and *D. sechellia*, respectively.

• Supplementary file 20. The summary of reads data used in this study.

• Supplementary file 21. The information and read coverage of each SNP in Y-linked gene families from Illumina reads. We listed the coverage of each SNP in Y-linked gene from each RNA-seq replicate and DNA-seq. We also recorded their frequency in our assembly and their translated amino acid. We estimated the expression level of each variant based on the SNP frequency in the genome. We also performed Welch's t-test to compare SNP frequency from DNA-seq and assemblies to it from RNA-seq. We further identify the SNPs associated with the allele that change more than 5 TPM compared to its estimated expression level from its frequency. The SNPs significant in the Welch's t-test and located in lowly or highly expressing alleles are chosen to perform the Chi-square test.

• Transparent reporting form

### Data availability

Genomic DNA sequence reads are in NCBI's SRA under BioProject PRJNA748438. All scripts and pipelines are available in GitHub (https://github.com/LarracuenteLab/simclade_Y; copy archived

at swh:1:rev:b1939db576cb1616094a59775a38014a7d61eb7f) and the Dryad digital repository (doi:https://doi.org/10.5061/dryad.280gb5mr6).

The following dataset was generated:

| Author(s) | Year | Dataset title | Dataset URL | Database and Identifier |
|---|---|---|---|---|
| Chang C, Gregory L, Gordon K, Meiklejohn C, Larracuente A | 2021 | Genome sequencing of males in *Drosophila* simulans clade | https://www.ncbi.nlm.nih.gov/bioproject/PRJNA748438 | NCBI BioProject, PRJNA748438 |
| Chang C, Gregory L, Gordon K, Meiklejohn CD, Larracuente A | 2021 | Unique structure and positive selection promote the rapid divergence of *Drosophila* Y chromosomes | https://doi.org/10.5061/dryad.280gb5mr6 | Dryad Digital Repository, 10.5061/dryad.280gb5mr6 |

The following previously published datasets were used:

| Author(s) | Year | Dataset title | Dataset URL | Database and Identifier |
|---|---|---|---|---|
| Garrigan et al. | 2012 | *Drosophila* mauritiana Genome sequencing | https://www.ncbi.nlm.nih.gov/bioproject/PRJNA158675 | NCBI BioProject, PRJNA158675 |
| Modencode S | 2012 | *D. melanogaster* Dissected Tissue RNASeq | https://trace.ncbi.nlm.nih.gov/Traces/sra/?study=SRP003905 | NCBI study, SRP003905 |
| Gerstein et al. | 2014 | modENCODE *D. melanogaster* Developmental Total RNA-Seq | https://trace.ncbi.nlm.nih.gov/Traces/sra/?study=SRP001696 | NCBI study, SRP001696 |
| Chakraborty et al. | 2017 | DSPR Founder Genomes | https://www.ncbi.nlm.nih.gov/bioproject/PRJNA418342/ | NCBI BioProject, PRJNA418342 |
| Wei et al. | 2018 | D. melanogaster, D. simulans, D. sechellia, D. erecta, D. ananassae, D. pseudoobscura, D. persimilis, D mojavensis, and D. virilis Raw sequence reads | https://www.ncbi.nlm.nih.gov/bioproject/PRJNA423291 | NCBI BioProject, PRJNA423291 |
| Laktionov et. al. | 2018 | Genome-wide profiling of gene expression and transcription factors binding reveals new insights into the mechanisms of gene regulation during *Drosophila* spermatogenesis [RNA-Seq] | https://www.ncbi.nlm.nih.gov/bioproject/PRJNA380909 | NCBI BioProject, PRJNA380909 |
| Lin et al. | 2018 | *Drosophila* simulans Raw sequence reads | https://www.ncbi.nlm.nih.gov/bioproject/PRJNA477366 | NCBI BioProject, PRJNA477366 |
| Shah et al. | 2020 | Novel quality metrics identify high-quality assemblies of piRNA clusters | https://www.ncbi.nlm.nih.gov/bioproject/PRJNA618654/ | NCBI BioProject, PRJNA618654 |
| Kim BY | 2021 | Nanopore-based assembly of many drosophilid genomes | https://www.ncbi.nlm.nih.gov/bioproject/PRJNA675888/ | NCBI BioProject, PRJNA675888 |
| Chakraborty et al. | 2021 | Transcriptome sequencing of *Drosophila* simulans clade | https://www.ncbi.nlm.nih.gov/bioproject/PRJNA541548 | NCBI BioProject, PRJNA541548 |

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

## Appendix 1

### Low Pacbio coverage in heterochromatic regions

We find that PacBio coverage is lower than expected on Y chromosomes and in heterochromatic regions generally (*Figure 1—figure supplement 2*). We found a similar bias in the *D. melanogaster* genome (*Chang and Larracuente, 2019*), where the PacBio data were independently generated by a different group (*Kim et al., 2014*). While a previous paper suggests that CsCl might contribute to this bias (*Krsticevic et al., 2015*), we used Qiagen's Blood and Cell culture DNA Midi Kit for DNA extraction. Heterochromatin is underreplicated in the endoreplicated cells that undergo multiple rounds of S phase but with no cell division such as those in larval salivary glands cells (*Smith and Orr-Weaver, 1991*). Previous studies demonstrated that endoreplicated cells in the adult flies might contribute to lower coverage in Illumina sequencing data (*Flynn et al., 2020*). Therefore, these endoreplicated cells might also contribute to the bias in Pacbio coverage.

### Validation of variants in Y-linked gene families

We mapped Illumina reads from male genomic DNA and testis RNAseq to the reconstructed ancestral transcript sequences of each gene cluster (*Lhk-1*, *Lhk-2*, *CK2ßtes-Y*) to estimate the expression level of the different Y-linked copies. We first asked if the variants in these two gene families found in our assemblies can be consistently detected in Illumina reads from male genomes. We found that the abundance of derived variants in these two gene families in the DNA-seq data are highly correlated to the frequency of variants in our assemblies ($R = 0.89$ and $0.98$ in *D. mauritiana* and *D. simulans,* respectively). For 559 variants in the *D. simulans* assembly, 33 of them (28 appear once and four appear twice) are missing from the DNA-seq data. For 446 variants in the *D. mauritiana* assembly, 43 of them (32 appear once and six appear twice) are missing from the DNA-seq data. Additionally, nine and eight inconsistent variants are located near ( < 100 bp) the start or end of transcripts in *D. simulans* and *D. mauritiana*, respectively. These regions at the edges of transcripts might have fewer Illumina reads coverage than more central regions.

We compared the proportion of synonymous and nonsynonymous changes between copies with high and low expression using transcriptome data to infer selection pressures on different mutations (*Figure 6—figure supplement 2*; *Supplementary file 21*).

To reduce the effect of sequencing errors and simplify the phylogenetic analyses on protein evolution rates, we first reconstructed the ancestral sequences of each gene cluster (*Lhk-1*, *Lhk-2*, *CK2ßtes-Y*, and 2 *CK2ßtes-like*; see *Figure 6*). The reconstructed ancestral sequences should eliminate misassembled bases, which are typically singletons. We conducted branch-model and branch-site-model tests on the reconstructed ancestral sequence using PAML and inferred that both gene families experienced strong positive selection following their duplication to the Y chromosome (from branch model; *Supplementary files 17 and 18*, *Figure 6*). The high rate of protein evolution in the Y-linked ampliconic genes suggests that, in addition to subfunctionalization or degeneration, they may also acquire new functions and adapt to being Y-linked.

