## [Editor Report]

This manuscript by Chang et al. reports the evolutionary patterns of Y-chromosome evolution in *Drosophila*, providing perhaps the most comprehensive interspecific comparison of Y chromosomes available to date. They focus on four species of the *melanogaster* species subgroup and do extensive sequencing and assembly. The manuscript describes the pattern of divergence in these chromosomes, and uses comparative approaches to compare the drivers of evolution in flies and mammals. The authors suggest that the Y chromosome uses a different mechanism to repair double strand breaks than on autosomes. We were impressed by the novelty and rigor of the work as well as the overall presentation of the results.

---

## [Decision Letter]

[Editors' note: this paper was reviewed by Review Commons.]

---

## [Author Response]

Reviewer #1 (Evidence, reproducibility and clarity (Required)):Summary:I found this an exceptionally impressive manuscript. The evolution of Y chromosomes has until recently been nearly impossible, and this research group have pioneered approaches that can yield reliable results in *Drosophila*. The study used an innovative heterochromatin-sensitive assembly pipeline on three D. simulans clade species, *D. simulans*, *D. mauritiana* and *D. sechellia*, which diverged less than 250 KYA, allowing comparisons with the group's previous results for the *D. melanogaster* Y.The study is both technically impressive and extremely interesting (an highly unusual combination). It includes a rich set of interesting results about these genome regions, and furthermore the results are discussed in a well-organised way, relating both to previous observations and to understanding of the genetics and evolution of Y chromosomes, illuminating all these aspects. It is a rare pleasure to read such a study. I believe that this study will inspire and be a model for future work on these chromosomes. It shows how these difficult genome regions can be studied.

Thank you for the positive evaluation of our paper. While we did not make any specific revisions in response to these comments, we did attempt to improve the writing.

Major comments:The conclusions are convincing. The methods are explained unusually clearly, and the reasoning from the results is convincing. When appropriate, the caveats, the caveats are clearly explained. The material is clearly organised and the questions studied are well related to the results. I had a few minor comments concerning the English. Even the figure (often a major problem to understand) are very clear and helpful, with proper explanations. I have very rarely read such a good manuscript, and almost never (in a long career) found a manuscript that could be published without revision being necessary.

Thank you for pointing out that there were minor concerns with the English. We have carefully gone through the manuscript and fixed some minor issues with the writing.

The analysis found 58 exons missed in previous assemblies (as well as all previously known exons of the 11 canonical Y-linked genes, which are present in at least one copy across the group). FISH on mitotic chromosomes using probes for 12 Y-linked sequences was used to determine the centromere locations, and to determine gene orders and relate them to the cytological chromosome bands, demonstrating changes in satellite distribution, gene order, and centromere positions between their Y chromosomes within the D. simulans clade species. It also confirmed previous results for Y-linked ribosomal DNA,genes, which are responsible for XY pairing in *D. melanogaster* males. Although 28S rDNA has been lost in D. simulans and *D. sechellia* (but not in D. mauritiana), the intergenic spacer (IGS) repeats between these repeats are retained on both sex chromosomes in all three species. Only sequencing can reliably reveal this, as their abundance is below the detection level by FISH in *D. sechellia*. The 11 canonical Y-linked genes' copy numbers vary between the species, and some duplicates are expressed and have complete open reading frames, and may therefore be functional because they, but most include only a subset of exons, often with duplicated exons flanking the presumed functional gene copy. Mega-introns and Y-loops were found, as already seen in *Drosophila* species, but this new study detects turn overs in the ~2 million years separating *D. melanogaster* and the *D. simulans* clade. 49 independent duplications onto the Y chromosome were detected, including 8 not previously detected. At least half show no expression in testes, or lack open reading frames, so they are probably pseudogenes. Testis-expressed genes may be especially likely to duplicate into the Y chromosome due to its open chromatin structure and transcriptional activity during spermatogenesis, and indeed most of the new Y-linked genes in the species studied clade have likely functions in chromatin modification, cell division, and sexual reproduction.The study discovered two new gene families that have undergone amplification on D. simulans clade Y chromosomes, reaching very high copy numbers (36-146). Both these families appear to encode functional protein-coding genes and show high expression. The paper described intriguing results that illuminate Y chromosome evolution.First, SRPK, arose by an autosome-to-Y duplication of the sequence encoding the testis-specific isoform of the gene SR Protein Kinase (SRPK), after which the autosomal copy lost its testis-specific exon via a deletion. In *D. melanogaster*, SRPK is essential for both male and female reproduction, so the relocation of the testis-specific isoform to the Y chromosome in the D. simulans clade suggests that the change may have been advantageous by resolving sexual antagonism. The paper presents convincing evidence that the Y copy evolved under positive selection, and that gene amplification may confer advantageous increased expression in males. The second amplified gene family is also potentially related to an interesting function. Both Xlinked and Y-linked duplicates are found of a gene called Ssl located on chromosome 2R. In D. simulans, the X-linked copies were previously known, and called CK2ßtes-like. In *D. melanogaster*, degenerated Y-linked copies are also found, with little or no expression, contrasting with complete open reading frames and high expression in the D. simulans clade species in testes, consistent with the possibility of an arms race between sex chromosome meiotic drive factors.Other interesting analyses document higher gene conversion rates compared to the other chromosomes, and evidence that these Y chromosomes may differ in the DNA-repair mechanisms (preferentially using MMEJ instead of NHEJ), perhaps contributing to their high rates of intrachromosomal duplication and structural rearrangements. The authors relate this to evidence for turnover of Y-linked satellite sequences, with the discovery of five new Y-linked satellites, whose locations were validated using FISH. The study also documented enrichment of LTR retrotransposons on the D. simulans clade Y chromosomes relative to the rest of the genome, together with turnovers between the species.Reviewer #1 (Significance (Required)):As described above, the advances are both, technical and conceptual for the field. The manuscript itself does an excellent job of placing the work in the context of the existing literature.Anyone working on sex chromosomes and other non-recombining genome regions should be interested in the findings reported.My field of expertise is the evolution of sex chromosomes, and the evolution of genome regions with suppressed recombination. I have experience of genomic analyses. I have less expertise in analyses of gene expression, but I understand enough about such approaches to evaluate the parts of this study that use them.Reviewer #2 (Evidence, reproducibility and clarity (Required)):The manuscript describes a thorough investigation of the Y-chromosomes of three very closely related *Drosophila* species (*D. simulans*, *D. sechellia*, and *D. mauritiana*) which in turn are closely related to *D. melanogaster*. The *D. melanogaster* Y was analysed in a previous paper by the same goup. The authors found an astonishing level of structural rearrangements (gene order, copy number, etc.), specially taking into account the short divergence time among the three species (~250 thousand years). They also suggest an explanation for this fast evolution: Y chromosome is haploid, and hence double-strand breaks cannot be repaired by homologous recombination. Instead, it must use the less precise mechanisms of NHEJ and MMEJ. They also provide circumstantial evidence that MMEJ (which is very prone to generate large rearrangements) is the preferred mechanism of repair. As far as I know this hypothesis is new, and fits nicely on the fast structural evolution described by the authors. Finally, the authors describe two intriguing Y-linked gene families in D. simulans (Lhk and CK2ßtes-Y), one of them similar to the Stellate / Suppressor of Stellate system of *D. melanogaster*, which seems to be evolving as part of a X-Y meiotic drive arms race. Overall, it is a very nice piece of work. I have four criticisms that, in my opinion, should be addressed before acceptance.

Thank you for your positive comments. We respond to your concerns point-by-point below.

The suggestion/conclusion that MMEJ is the preferential repair mechanism (over NHEJ) should be better supported and explained. At line 387, the authors stated "The pattern of excess large deletions is shared in the three D. simulans clade species Y chromosomes, but is not obvious in *D. melanogaster* (Figure 6B). However, because all *D. melanogaster* Y-linked indels in our analyses are from copies of a single pseudogene (CR43975), it is difficult to compare to the larger samples in the simulans clade species (duplicates from 16 genes). ". Given that *D. melanogaster* has many Y-linked pseudogenes (described by the authors and by other researchers, and listed in Table S6), there seems to be no reason to use a sample size of 1 in this species.

We only used pseudogenes with large alignable regions (>300 bp) to prevent the potential bias toward small indels and increase our confidence in indel calling. As a result, we excluded most of the duplicates on the *D. melanogaster* Y chromosome*.* We now include 5 additional *D. melanogaster* Y-linked indels in the manuscript, however, the majority of indels in this species (36/41) are still from the same gene.

Furthermore, given that *D. melanogaster* is THE model organism, it is the species that most likely will provide information to assess the "preferential MMEJ" hypothesis proposed by the authors.

A previous paper has shown that male flies deficient in MMEJ have a strong bias toward female offspring (McKee et al. 2000), suggesting that MMEJ is necessary for successfully producing Y-bearing sperm, consistent with our hypothesis. We agree with the reviewer that careful genetic and cytological experiments in *D. melanogaster* could further clarify the role of MMEJ in the repair of Y-linked mutations. Even more revealing would be experiments using the simulans clade species, where we hypothesize the MMEJ bias is even more pronounced on the Y chromosome. We believe, however, that these experiments are beyond the scope of this study and should merit their own papers.

Still on the suggestion/conclusion that MMEJ is the preferential repair mechanism (over NHEJ). Y chromosome in heterochromatic, haploid and non-recombining. In order to ascribe its mutational pattern to the haploid state (and the consequent impossibility of homologous recombination repair), the authors compared it to chromosome IV (the so called "dot chromosome"). This may not be the best choice: while chr IV lacks recombination in wild type flies, it is not typical heterochromatin. E.g., " results from genetic analyses, genomic studies, and biochemical investigations have revealed the dot chromosome to be unique, having a mixture of characteristics of euchromatin and of constitutive heterochromatin". Riddle and Elgin, FlyBook 2018 (https://doi.org/10.1534/genetics.118.301146). Given this, it seems appropriate to also compare the Y-linked pseudogenes with those from typical heterochromatin. In *Drosophila*, these are the regions around the centromeres ("centric heterochromatin"). There are pseudogenes there; e.g., the gene rolled is known to have partially duplicated exons.

Thank you for the suggestion. We now include the data from pericentric heterochromatin and pseudogenes in supplemental data (see Figure 7). Both data types support our conclusion that indel size is only larger on Y chromosomes, which is consistent with the comparison between the dot chromosome and pericentric heterochromatin reported by Blumenstiel et al. 2002.

In some passages of the manuscript there seems to be a confusion between new genes and pseudogenes, which should be corrected. For example, in line 261: "Most new Y-linked genes in *D. melanogaster* and the D. simulans clade have presumed functions in chromatin modification, cell division, and sexual reproduction (Table S7)".. Who are these "new genes"? If they are those listed in Table S6 (as other passages of the text suggest), most if not all of them are pseudogenes. If they are pseudogenes, it is not appropriate to refer to them as "new genes". The same ambiguity is present in line 263: "Y-linked duplicates of genes with these functions may be selectively beneficial, but a duplication bias could also contribute to this enrichment (…) " Pseudogenes can be selectively beneficial, but in very special cases (e.g.. gene regulation). If the authors are suggesting this, they must openly state this, and explain why. Pseudogenes are common in nearly all genomes, and should be clearly separated from genes (the later as a shortcut for functional genes). The bar for "genes" is much higher than simple sequence similarity, including expression, evidences of purifying selecion, etc., as the authors themselves applied for the two gene families they identified in D. simulans (Lhk and CK2ßtes-Y).

Thank you for the suggestion. We now state our criteria for calling genes based on the expression and long CDS and correct the sentences that the reviewer refers to. The protein evolution rates of many Y-linked duplicates were surveyed in Tobler et al. 2017, who found that most are not under strong purifying selection. Our study supports this previous report. We think that protein evolution rate alone may not be a good indicator for functionality. Our current study does not focus on the potential function of these genes, and we think further population studies are required to get a solid conclusion. We changed the text to clarify this point: “Most new Y-linked duplications in *D. melanogaster* and the *D. simulans* clade are from genes with presumed functions in chromatin modification, cell division, and sexual reproduction (Supplementary file 8, formerly Table S7), consistent with other *Drosophila* species [17, 77].” (p15 L281-284).

The authors center their analysis on "11 canonical Y-linked genes conserved across the melanogaster group ". Why did they exclude the CG41561 gene, identified by Mahajan and Bachtrog (2017) in *D. melanogaster?* Given that most *D. melanogaster* Y-linked genes were acquired before the split from the D. simulans clade (Koerich et al. Nature 2008), the same most likely is true for CG41561 (i.e., it would be Y-linked in the D. simulans clade). Indeed, computational analysis gave a strong signal of Y-linkage in D. yakuba (unpublished; I have not looked in the other species). If CG41561 is Y-linked in the simulans clade, it should be included in the present paper, for the only difference between it and the remaining "canonical genes" was that it was found later. Finally, the proper citation of the "11 canonical Y-linked genes" is Gepner and Hays PNAS 1993 and Carvalho, Koerich and Clark TIG 2009 (or the primary papers), instead of ref #55.

Thank you for the suggestion. *CG41561* is indeed a relatively young Y-linked gene because it’s not Y-linked in *D. ananassae* (Muller’s element E). We already have *CG41561* in Supplementary file 7 (formerly Table S6) and we think that it is reasonable to separate a young Y-linked gene from the others. We also fixed the reference as suggested (p5 L116).

Other points/comments/suggestions:a. Possible reference mistake: line 88 "For example, 20-40% of *D. melanogaster* Y-linked regulatory variation (YRV) comes from differences in ribosomal DNA (rDNA) copy numbers [52, 53]." reference #53 is a mouse study, not *Drosophila*.

Thank you for pointing out this error, we fixed the reference (p4 L91).

b. Possible reference mistake: line 208 "and the genes/introns that produce Y-loops differs among species [75]". ref #75 is a paper on the D. pseudoobscura Y. Is it what the authors intended?

Yes, our previous paper (ref 75) found that Y-loops do not originate from the kl-3, kl-5, and ORY genes in *D. pseudoobscura* because they don’t have large introns in this species.

c. line 113. "We recovered all known exons of the 11 canonical Y-linked genes conserved across the melanogaster group, including 58 exons missed in previous assemblies (Table S1; [55])." Please show in the Table S1 which exons were missing in the previous assemblies. I guess that most if not all of these missing exons are duplicate exons (and many are likely to be pseudogenes). If they indeed are duplicate exons, the authors should made it clear in the main text, e.g., "We recovered all known exons of the 11 canonical Y-linked genes conserved across the melanogaster group, plus 58 duplicated exons missed in previous assemblies."

Thank you for the suggestion. However, the 58 exons did not include the duplicated exons. We are similarly surprised how much we will miss if we don’t assemble the Y chromosome carefully. We now mark these exons in red in Supplementary file 1 (formerly Table S1) to make this point clearer.

d. line 116 "Based on the median male-to-female coverage [22], we assigned 13.7 to 18.9 Mb of Y-linked sequences per species with N50 ranging from 0.6 to 1.2 Mb." The method (or a very similar one) was developed by Hall et al. BMC Genomics 2013, which should be cited in this context. (e) line 118: "We evaluated our methods by comparing our assignments for every 10-kb window of assembled sequences to its known chromosomal location. Our assignments have 96, 98, and 99% sensitivity and 5, 0, and 3% false-positive rates in *D. mauritiana*, *D. simulans*, and *D. sechellia*, respectively (Table S2). The procedure is unclear. Why break the contigs in 10kb intervals, instead of treating each as an unity, assignable to Y, X or A? The later is the usual procedure in computational identification of suspect Y-linked contigs (Carvalho and lark Gen Res 2013; Hall et al. BMC Genomics 2013). The only reason I can think for analyzing the contigs piecewise is a suspicion of misassemblies. If this is the case, I think it is better to explain.

Thank you for the suggestion. We did not break the contigs into 10kb intervals when we assigned the Y-linked contigs. As you suspect, our motivation for evaluating our methods and analyzing the contigs in 10kb intervals was to detect possible misassemblies. We rewrote the sentence to make this point clearer (p6 L129-132).

e. Figure 1. It may be interesting to put a version of Figure 1 in the SI containing only the genes and the lines connecting them among species, so we can better see the inversions etc. (like the cover of Genetics , based on the paper by Schaeffer et al. 2008).

Thank you for the suggestion. We would like to make a figure like that fantastic cover image you refer to, but the repetitive nature of the Y chromosome makes it difficult to illustrate rearrangements based on alignments at the contig-level. We instead opted to update Figure 1 to better highlight the rearrangements, still based on the unique protein-coding genes which are supported by the FISH experiments.

f. Table S6 (Y-linked pseudogenes). Several pseudogenes listed as new have been studied in detail before: vig2, Mocs2, Clbn, Bili (Carvalho et al. PNAS2015) Pka-R1, CG3618, Mst77F (Russel and Kaiser Genetics 1993; Krsticevic et al. G3 2015). Note also that at least two are functional (the vig2 duplication and some Mst77 duplications).

Thank you for the suggestion. We now include a column to indicate the potential function of Y-linked duplicates (see Supplementary file 7, formerly Table S6).

g. line 421: "one new satellite, (AAACAT)n, originated from a DM412B transposable element, which has three tandem copies of AAACAT in its long terminal repeats." The birth of satellites from TEs has been observed before, and should be cited here. Dias et al. GBE 6: 1302-1313, 2014.

Thank you for the suggestion. We now include a sentence to cite this reference (p27 L467-468).

h. Figure S2 shows that the coverage of PacBio reads is smaller than expected on the Y chromosome. Any explanation? This has been noticed before in *D. melanogaster*, and tentatively attributed to the CsCl gradient used in the DNA purification (Carvalho et al. GenRes 2016). However, it seems that the CsCl DNA purification method was not used in the simulans clade species (is it correct?). Please explain the in the manuscript, or in the SI. The issue is relevant because PacBio sequencing is widely believed to be unbiased in relation to DNA sequence composition (e.g., Ross et al. Genome Biol 2013).

Yes, we used Qiagen's Blood and Cell Culture DNA Midi Kit for DNA extraction. We suspect that the underrepresentation of Y-linked reads is driven by the presence of endoreplicated tissue in adults. Heterochromatin is underreplicated in endoreplicated cells, and thus there may simply be less heterochromatin in these tissues. Consistent with this idea, we find that all heterochromatin seems to be underrepresented in the reads, not just the Y chromosome (see Chakraborty et al. 2021; Flynn et al. 2020). We now include this discussion in the SI of our paper (see supplementary text p75).

i. I may have missed it, but in which public repository have the assemblies been deposited?

We link to the assemblies in Github (https://github.com/LarracuenteLab/simclade_Y) and they will also be in the Dryad Digital Repository (doi forthcoming).

Reviewer #3 (Evidence, reproducibility and clarity (Required)):Due to suppressed recombination, Y chromosomes have degenerated, undergone extensive structural rearrangements, and accumulated ampliconic gene families across species. The molecular processes and selective pressures guiding dynamic Y chromosome evolution are not well understood. In this study, Chang et al. generate updated Y assemblies of three closely related species in the D. simulans complex using long-read PacBio sequencing in combination with FISH. Despite having diverged only 250,00 years ago, the authors find structural rearrangements, two newly amplified gene families and evidence of positive selection across D. simulans. The authors also suggest the high level of Y duplications and deletions may be mediated by MMEJ biased repair.

Our aim is to discover and understand the many different factors and processes that shape the evolution of Y chromosome organization and function. Because these Y chromosomes were largely unassembled, we needed to first generate the sequence assembly before we could ask specific questions. We prefer not to focus the manuscript solely on one specific topic such as MMEJ repair, as our other observations and analyses may be interesting to a wide range of scientists studying topics other than mutation and DNA repair. We are therefore choosing to present the more comprehensive story about Y chromosome evolution that we included in our original manuscript.

We also respectfully disagree with the comment that our paper is just a descriptive survey of Y chromosomal sequence features. On the contrary, we present thorough evolutionary analyses to test hypotheses about the forces shaping the evolution of Y chromosome organization and Y-linked genes. Specifically, we use molecular evolution and phylogenetic and comparative genomics approaches to show that multi-copy gene families experience rampant gene conversion and positive selection. We posit that one simulans clade-specific Y-linked gene family has undergone subfunctionalization, potentially resolving sexual conflict, and another may be involved in meiotic drive. We also use evolutionary genomic approaches to show that the distribution of Y-linked mutations indeed suggests that Y chromosomes disproportionately use MMEJ and we propose that this unique feature may shape the evolution of Y chromosome structural organization. This is, as far as we know, a novel hypothesis. We think that follow-up studies of either hypothesis merit different papers.

Major concerns:1. Title: The authors use "unique structure" in the title, which is a vague point. Are not Y chromosomes, or any chromosome, "unique" in some manner? Also are there not more evolutionary processes governing the rapid divergence of the Y's.

Thank you for raising your concern. We believe that we are justified in referring to the Y chromosome as unique among all other chromosomes in its structural properties (e.g. combination of its hemizygosity, abundant tandem repeats, large scale rearrangements, and highly amplified testis-specific genes). Because there are many properties of Y chromosomes that we believe contribute to their rapid divergence, we opted for the general phrase ‘unique structure’ to capture all of these features. Many evolutionary processes likely shape the evolution of that unique structure (e.g. Muller’s Ratchet, background selection, Hill Robertson effects; see Charlesworth and Charlesworth 2000 for a review), and these processes are well-studied, especially on newly evolved sex chromosomes. Here our focus is on evolutionarily old Y chromosomes, which may have comparatively fewer targets of purifying selection and are more likely to be shaped by positive selection (Bachtrog 2008).

2. p.2, line 53-56: The authors claim that sexually antagonistic selection and regulatory evolution are causes of recombination suppression. Couldn't this statement be reversed? Recombination suppression via inversions or other rearrangements enable sexually antagonistic selection. This is a chicken or egg question, so it should be revised to have both possibilities be equal.

Thank you for the suggestion. We think that it is unlikely that recombination suppression itself is beneficial, but for sexually antagonistic selection and regulatory evolution, recombination suppression can have short-term benefits. We rephrased this sentence to be agnostic about the direction (p2 L56).

3. p.5, 118-120: Are the assemblies de novo or have they been guided based upon the D.melanogaster Y chromosome assembly? Please clarify how the authors evaluate their methods by comparing their Y-sequence assignments to known chromosomal locations.

Thank you for the suggestion. We didn’t use *D. melanogaster* Y chromosome assembly to guide our assemblies. “All assemblies are generated de novo”, and thus we don’t think there is any potential bias. We first assigned Y-linked sequences using the presence of known Y-linked genes, and used this assignment to evaluate our methods. We now make the sentence clear (p5 L112).

4. While the gene copy number estimates are accurate, the PacBio-based genome assemblies are still not able to accurately assemble large segmental duplications (see Evan Eichler's laboratories recent primate and human genome assemblies). A statement mentioning the concerns about accuracy of the underlying sequence and genomic architecture shown should be included in the main text. FISH provides support for the location of the contigs, but not for the accuracy of the underlying genomic architecture.

Thank you for the suggestion. We can’t validate all Y-linked regions. We did validate the larger structural features of the assembly and only discuss the results that we are confident in. We now include sentences to address this concern (p7 L150-152).

5. The authors assigned Y-linked sequences based on median male-to-female coverage. Is this method feasible for assigning ampliconic sequence to the Y given the N50 of 0.6-1.2Mb? Are the authors potentially excluding novel Y-linked ampliconic sequence?

We validated our methods to assign contigs to a chromosome by comparing 10-kb intervals to the contigs with known chromosomal location, including the Y chromosome. Our assignments have high (96, 98, and 99%) sensitivity and low (5, 0, and 3%) falsepositive rates in *D. mauritiana*, *D. simulans*, and *D. sechellia*, respectively (see Supplementary file 2, formerly Table S2). Based on these results, we think that this method is reasonable for Y-linked contigs with N50 of 0.6-1.2Mb.

We might exclude some novel Y-linked sequences since we only assigned ~15Mb out of a total ~40 Mb Y-linked sequences. We acknowledged this possibility, and now include a sentence to address this concern (p31 L554-556).

6. Where did the rDNA sequences go in *D. simulans* and *D. sechellia?* Can they be detected on another chromosome?

Please see Figure S5 for detailed results. We found a few copies of rDNA on the contigs of autosomes. We assembled many copies of rDNA that can’t be confidently assigned to Y chromosomes. It’s possible that they might be located on other chromosomes. Based on our FISH data (Figure S4) and previous papers, most of these non-Y-linked rDNA copies should be on the X chromosome. However, in this study, we did not make a concerted effort to assign X-linked contigs.

7. Figure 2B is hard to follow and it is unclear what additional value it provides to part A. Why is expression level of specific exons important?

Exon duplication may be an important contributor to Y-linked gene evolution: most genes have duplications and our figure shows that at least some of these duplicates are expressed. The patterns we see indicate that duplication may play different roles in genes depending on their length. For example, the duplications involving short genes (e.g., ARY) may be functional and influence protein expression, whereas duplications involving large genes (e.g. kl-2) may not influence the overall protein expression level from this gene, although the expressed duplicated exons may play some other role. We revised a sentence in the main text and added a sentence to the figure 2 legend to make this point clearer.

8. Figure 3 There are many introns that contain gaps, so it is unclear how confident one can be in intron length when there are gaps.

Indeed, we are not confident about the length of introns with gaps. Therefore, we separated these introns and showed them in different colors.

9. Figure 4: What are the authors using as a common ancestor in this figure to infer duplications in the initial branch?

We used phylogenies to infer the origin of Y-linked duplicates. Any duplications that happened earlier than the divergence between four species are listed in the branch. We also edited the legend to make this point clearer.

10. p.15, paragraph 2: The authors describe a newly amplified gene, CK2Btes-Y, in D. simulans. In the first half of the paragraph the authors state that Y-linked copies are also found in *D. melanogaster* but have "degenerated and have little or no expression" and call them pseudogenes. Later in the paragraph, the authors state that the *D. melanogaster* Y-linked copies are Su(Ste), a source of piRNAs that are in conflict with X-linked Stellate. Lastly in the paragraph, the authors discuss Su(ste) as a *D. melanogaster* homolog of CK2Btes-Y. The logic of defining CK2Btes-Y origins is confusing. Was CK2Btes-Y independently amplified on the D. simulans Y, or were CK2BtesY and Su(Ste) amplified in a common ancestor but independently diverged?

The amplification of *CK2Btes-Y* and *CK2Btes-like* happened in the ancestor of *D.*

melanogaster and D. simulans (Figure S11). However, both CK2Btes-Y and CK2Btes-like became pseudogenes (*D. melanogaster* CK2Btes-Y is named PCKR in a previous study) in *D. melanogaster*. On the other hand, Ste and Su(Ste) are only limited to D.

*melanogaster* based on phylogenetic analyses (Figure 5A) and are a chimera of *CK2Bteslike* and *NACBtes*. The evolutionary history of this gene family has been detailed in other papers, except for the presence of *CK2Btes-Y* in the *D. simulans* complex, which we describe for the first time in this study. We now include a new figure (Figure 5B) a schematic of the inferred evolutionary history of sex-linked *Ssl/CK2ßtes* paralogs

11. Figure 5: Is each FISH signal a different gene copy?

Yes, based on our assemblies, *Lhk-1* and *Lhk-2* are mostly located on different contigs. Unfortunately, we are not able to design probes that can separate *Lhk-1* from *Lhk-2*.

12. The authors suggest DNA-repair on the Y chromosome is biased towards MMEJ based on indel size and microhomologies. Is there any evidence MMEJ is responsible for variable intron length in the canonical Y-linked genes or the amplification of new gene families? Since MMEJ is error-prone, it's a more tolerable repair mechanism in pseudogenes, so their findings might be biased. Rather than comparing pseudogenes to their parent genes, they should compare chrY pseudogenes to autosomal pseudogenes. Even more would be to track MMEJ on the dot chromosome which is known not recombine and is highly heterchromatic like the Y chromosome.

We did compare chrY pseudogenes to autosomal pseudogenes in our study. We also add new analyses to address other issues from reviewer 2, which are similar to your concern. We now include data from pericentric heterochromatin and pseudogenes (see Figure 7). Both data types support our conclusion that indel size is only larger on Y chromosomes. This is consistent with a report that the dot chromosome and pericentric heterochromatin have similar indel size distributions (Blumenstiel et al. 2002).

Reviewer #3 (Significance (Required)):While it is a benefit to have much improved Y chromosome assemblies from the three D. simulans clade species, the gap in knowledge this manuscript is trying to address is unclear.The manuscript is almost entirely descriptive and the figures are difficult to follow.

As stated above, we respectfully disagree with the comment that the manuscript is entirely descriptive, as we present thorough evolutionary analyses to test hypotheses about the forces shaping the evolution of Y chromosome organization and Y-linked genes. We have two guiding hypotheses about the importance of sexual antagonism and DNA repair pathways for Y chromosome evolution, and we conduct sequence analyses that support these hypotheses that sexual antagonism and MMEJ affect Y chromosome evolution.

References:

Bachtrog D. The temporal dynamics of processes underlying Y chromosome degeneration. Genetics. 2008 Jul;179(3):1513-25. doi: 10.1534/genetics.107.084012. Epub 2008 Jun 18. PMID: 18562655; PMCID: PMC2475751.

Blumenstiel, J.P., Hartl, D.L, Lozovsky, E.R.. Patterns of Insertion and Deletion in

Contrasting Chromatin Domains, Molecular Biology and Evolution, Volume 19, Issue 12,

December 2002, Pages 2211–2225, https://doi.org/10.1093/oxfordjournals.molbev.a004045

Chakraborty M, Chang CH, Khost DE, Vedanayagam J, Adrion JR, Liao Y, Montooth KL, Meiklejohn CD, Larracuente AM, Emerson JJ. Evolution of genome structure in the *Drosophila* simulans species complex. Genome Res. 2021 Mar;31(3):380-396. doi:

10.1101/gr.263442.120. Epub 2021 Feb 9. PMID: 33563718; PMCID: PMC7919458.

Charlesworth B, Charlesworth D. The degeneration of Y chromosomes. Philos Trans R Soc Lond B Biol Sci. 2000 Nov 29;355(1403):1563-72. doi: 10.1098/rstb.2000.0717. PMID: 11127901; PMCID: PMC1692900.

Flynn,J, Long, M, Wing, RA, A.G Clark, Evolutionary Dynamics of Abundant 7-bp

Satellites in the Genome of *Drosophila* virilis, Molecular Biology and Evolution, Volume 37, Issue 5, May 2020, Pages 1362–1375, https://doi.org/10.1093/molbev/msaa010

McKee, Bruce D. et al. “On the Roles of Heterochromatin and Euchromatin in Meiosis in *Drosophila*: Mapping Chromosomal Pairing Sites and Testing Candidate Mutations for Effects on X–Y Nondisjunction and Meiotic Drive in Male Meiosis.” Genetica 109 (2004): 77-93.

Tobler R, Nolte V, Schlötterer C. High rate of translocation-based gene birth on the *Drosophila* Y chromosome. Proc Natl Acad Sci U S A. 2017 Oct 31;114(44):11721-11726. doi: 10.1073/pnas.1706502114. Epub 2017 Oct 19. PMI